# Synthesis of New Star-Shaped Liquid Crystalline Cyclotriphosphazene Derivatives with Fire Retardancy Bearing Amide-Azo and Azo-Azo Linking Units

**DOI:** 10.3390/ijms21124267

**Published:** 2020-06-16

**Authors:** Zuhair Jamain, Melati Khairuddean, Tay Guan-Seng

**Affiliations:** 1Faculty of Science and Natural Resources, Universiti Malaysia Sabah (UMS), Kota Kinabalu 88400, Malaysia; 2School of Chemical Sciences, Universiti Sains Malaysia (USM), Penang 11800, Malaysia; 3School of Industrial Technology, Universiti Sains Malaysia (USM), Penang 11800, Malaysia; taygs@usm.my

**Keywords:** cyclotriphosphazene, amide-azo, azo-azo, liquid crystal, fire retardancy

## Abstract

Two series of new hexasubstituted cyclotriphosphazene derivatives were successfully synthesized and characterized. These derivatives are differentiated by two types of linking units in the molecules such as amide-azo (**6a–j**) and azo-azo (**8a–j**). The homologues of the same series contain different terminal substituents such as heptyl, nonyl, decyl, dodecyl, tetradecyl, hydroxyl, carboxyl, chloro, nitro, and amino groups. All the intermediates and final compounds were characterized using Fourier transform infrared spectroscopy (FT-IR), nuclear magnetic resonance spectroscopy (NMR), and Carbon, Hydrogen, and Nitrogen (CHN) elemental analysis. Liquid crystal properties for all compounds were determined using polarized optical microscope (POM). It was found that only intermediates **2a–e** with nitro and alkoxyl terminal chains showed a smectic A phase. All the final compounds with alkoxyl substituents are mesogenic with either smectic A or C phases. However, other intermediates and compounds were found to be non-mesogenic. The study on the fire retardancy of final compounds was determined using limiting oxygen index (LOI) method. The LOI value of pure polyester resin (22.53%) was increased up to 24.71% after treating with 1 wt% of hexachlorocyclotriphosphazene (HCCP). Moreover, all the compounds gave positive results on the LOI values and compound **6i** with the nitro terminal substituent showed the highest LOI value of 27.54%.

## 1. Introduction

The extensive exploration of phosphazene liquid crystalline materials having flexible ordered structures has led to the discovery of new properties and applications, such as advanced technological devices or smart biological and pharmacological uses [1]. The latest development of liquid crystal materials has demonstrated that these compounds are able to detect small changes in temperature, electromagnetic radiation, mechanical stress, and chemical environment [2,3,4].

Liquid crystal (LC) is an intermediate phase, that is, a matter that has both properties of crystal solid and isotropic liquid [5,6,7]. In a crystal solid, the molecules are arranged in a highly ordered system having the positional and orientational order. As the temperature increases, the solid absorbs some heat and melts into a liquid crystal phase whereby the molecules are able to move freely, disrupting the positional order without changing the orientational order [8,9]. The various LC phases can differ in the type of order, whereby in positional order, the molecules are arranged in any sort of ordered lattice, while in orientational order, molecules are pointing in the same direction, either short-range (between molecules close to each other) or long-range (extending to larger dimensions). However, with further heating, the molecules lose both the positional and orientational order whereby they are free to move and randomly distributed [10]

Calamitic or rod-like liquid crystals are compounds with an elongated shape in which the molecular length is greater than breadth. The molecules possess strong attractive forces and tend to point in the same direction which helps to stabilize the structure of the compound [11]. A common structural feature of calamitic LC consists of two or more aromatic rings, which are connected to the side chain or terminal group by means of linking units. Discotic LC or disc-like molecules, on the other hand, possess a rigid core surrounded by commonly six side arms of the calamitic molecules [12]. When the molecules are oriented themselves in a layer-like fashion, a discotic nematic phase can be observed, but when the molecules are packed into stacks, the columnar phase(s) will be seen. The structural representation is shown in Figure 1.

Hexachlorocyclotriphosphazene, HCCP, is an aromatic ring compound consisting of alternating phosphorus and nitrogen atoms with two chlorine atoms attached to each phosphorus atom. The multiarmed rigid structure of HCCP caused the exploration of the discotic structures and their application as liquid crystal materials was investigated. Allcock and Klingenberg have described the aromatic azo phosphazene polymer liquid crystals. They have investigated the influence of the substituents on the aromatic unit such as the function of the chain length and different phosphazene skeletal structures [13]. Moriya and coworkers reported interesting liquid crystal properties of the organophosphazene backbones decorated by mesogenic side chains [14,15,16,17,18,19,20,21,22]. Meanwhile, the calamitic phases were exhibited when the calamitic units are linked to the phosphazene ring [23,24,25]. It was also reported that liquid crystal phosphazene with aryloxy side groups can act as excellent flame retardants and proved to provide higher thermal stabilities to the system [26]. In addition, cyclotriphosphazenes with alkyl chain increased the thermal properties and fire retardancy because of the phosphorous and nitrogen flame-retardant synergy [27,28].

In order to avoid complications from varieties of end group associations, the main variables that will be studied are the effect of the chain length, the type of end substituents, and the influence of the cyclotriphosphazene skeletal structures towards the liquid crystal properties. Nevertheless, to our best knowledge, only few kinds of cyclotriphosphazene liquid crystals compounds with two linking units have been studied. Our interest is to gain a better insight into the structure–properties relationship of these types of compounds. The addition of cyclotriphosphazene to a compound aims to increase the resistance of the material towards ignition. The fire-retardant function is to minimize the fire risk and to prevent a small fire from becoming a major catastrophe. Shuang et al. (2016) reported the preparation of hexa(4-maleimido-phenoxyl) cyclotriphosphazene and proved its possible application as a fire retardant [29].

Even though the substituted cyclotriphosphazene derivatives were extensively studied as fire retardants, the investigation of the cyclophosphazene-based compounds having organic side arms and possessing the liquid crystal and fire retardant behaviour is still not being fully explored. Ordinarily, the chlorine atoms in the HCCP are replaced with organic side arm in order to form halogen-free fire-retardant compounds. The traditional halogen-containing fire retardants is forbidden due to its toxicity and safety awareness [30]. Interestingly, the incorporation of hexachlorocyclotriphosphazene into a compound is to increase the resistance of the material towards ignition. Thus, this research focused on the preparation of a new series of hexasubstituted cyclotriphosphazene derivatives with amide and azo (Series 1) and two azo (Series 2) bridging units. Extension of the linking unit greatly enhances both the LC phases and the thermal stability of the compounds. 

## 2. Results and Discussion

### 2.1. Synthesis of the Intermediates and Final Compounds

Diazotization reaction of *p*-nitroaniline with phenol forms 4-(4-nitrophenylazo)phenol, **1**, which is alkylated with heptyl, nonyl, decyl, dodecyl, or tetradecyl bromide to give a series of nitro intermediates, **2a–e**. Reduction of **2a–e** or **1** gives the subsequent amine, **3a–f**. Another similar reaction with a series of substituted aniline results in **3g–i**. As a result of the reaction of hexachlorocyclotriphosphazene, HCCP, with methyl 4-hydroxybenzoate, the hexasubstituted cyclotriphosphazene benzoate, **4** is formed, which was then hydrolyzed to give the subsequent benzoic acid, **5**. Further reaction of **5** with **3a–i** produces the first series of hexasubstituted cyclotriphosphazene derivatives with amide and azo linking units, **6a–i**. Meanwhile, compound **6j** is formed by means of the reduction of **6i**. Another reaction of HCCP with 4-nitrophenol gives intermediate **7**, which further reacts with **3a–i** to form the second series of hexasubstituted cyclotriphosphazene derivatives with two azo linking units, **8a–i**. Intermediate **9** was produced from the reaction of HCCP with 4-acetamidophenol, and then deprotected to form intermediate **10**. Lastly, compound **8j** was synthesized by the reaction of intermediate **10** with **3i** with amino terminal end. The general synthesis method for these intermediates and final compounds are described in Scheme 1, Scheme 2, Scheme 3 and Scheme 4.

In this research, all the synthesized intermediates and compounds were characterized using FTIR spectroscopy, NMR spectroscopy, and CHN elemental analysis. The texture of these compounds was determined using POM and their mesophase transition were further confirmed using DSC. The fire-retardant properties of the compounds were investigated using LOI testing. The full data of the intermediates and final compounds can be found in Section 3.3. 

### 2.2. FTIR Spectral Discussion

The FTIR data of intermediate **1** showed a broad band at 3250 cm^−1^ for the O-H stretching. The alkylation of **1** formed **2a–e** whereby the band at 3250 cm^−1^ disappeared and two bands at 2855 and 2920 cm^−1^ (C_sp3_-H stretching) were seen. The reduction of **1** and **2a–e** formed **3a–f** was successful as evident from the appearance of two spikes at 3335 and 3470 cm^−1^ due to the N-H stretching. Other absorption bands of intermediates **1**, **2a–e**, and **3a–i** were consistent with the spectral data. The hexasubstituted cyclotriphosphazene benzoate, **4** showed the absorption at 2875 and 2930 cm^−1^ (C_sp3_-H stretching), indicating the presence of the methyl group in the side arms. Hydrolysis of hexasubstituted cyclotriphosphazene benzoate, **4** leads to intermediate **5** (an acid) which confirmed by the appearance of a broad band at 3250 cm^−1^ (O-H stretching). Intermediate **5** was reacted with thionyl chloride to form an acid chloride. The acid chloride was further utilized with **3a–i** to form the final compounds, **6a–i**, with amide and azo linking units (Series 1).

Based on the FTIR spectra overlay of compounds **6a–i** (Figure 2), the appearance of the N-H stretching of amide group at 3343 cm^−1^ confirms the successful reactions of the acid chloride with intermediates **3a–i** (Scheme 2). 

Compounds **6a–e** with the alkoxy substituents showed the absorption bands for the symmetrical and asymmetrical C_sp3_-H stretching at 2850 and 2920 cm^−1^. The presence of a broad signal for O-H stretching in IR spectra of compounds **3f** and **3g** is due to the hydroxyl and carboxyl substituents, respectively. Moreover, the C-Cl bending of compound **3h** is located at 811 cm^−1^ and the azo linkages (N=N stretching) absorbed at 1485 cm^−1^. The reduction of compound **6i** to **6j** leads to the appearance of the absorption band at 3218 and 3381 cm^−1^ for the amino group (N-H stretching). These N-H stretching represents a broad absorption band which might due to the presence of moisture from the sample or surroundings.

Another hexasubstituted cyclotriphosphazene intermediate, **7** was successfully synthesized from the reaction between HCCP and 4-nitrophenol. The disappearance band at 3320 cm^−1^ for O-H stretching indicated that 4-nitrophenol was successfully attached to HCCP. This intermediate was then reacted with intermediates **3a****–i** to form compounds **8a****–i** with two azo linking units (Series 2). Moreover, compound **8j** was synthesized by a different synthetic route. The substitution reaction of HCCP with 4-acetamidophenol in a basic solution yielded hexasubstituted intermediate, **9**, which was deprotected to give intermediate **10** bearing an amino group in the side arms. The FTIR spectrum of **9** showed a strong absorption at 3343 (N-H stretching) and 1700 cm^−1^ (C=O stretching). The disappearance of these two absorptions in intermediate **10**, and the appearance of two absorption spikes at 3217 and 3451 cm^−1^ for the N-H stretching confirmed that the deprotection reaction was successful. Intermediate **10** was reacted with **3i** to form compound **8j** with the absorption bands at 3217 and 3451 cm^−1^, confirming that the coupling reactions was successful.

According to the FTIR overlay spectra of compounds **8a–j** (Figure 3), the N=N stretching was assigned at 1493 cm^−1^. The disappearance of the N-H stretching from the starting materials confirmed the formation of N=N bond in the compounds. The alkoxy chains in compounds **8a–e** gave the absorptions at 2860 and 2921 cm^−1^ for the symmetrical and asymmetrical C_sp3_-H stretching. Meanwhile, the hydroxyl and carboxyl groups in compounds **8f** and **8g**, respectively, displayed the broad absorption at 3255 cm^−1^ for the O-H stretching. As shown in the other series, compound **8h** with chlorine substituent showed the C-Cl bending at 790 cm^−1^ while the P=N stretching and P-O-C bending of cyclotriphosphazene ring were observed at 1189 and 997 cm^−1^, respectively. Only compound **8j** showed the N-H stretching which due to the amino (NH_2_) group as a substituent at the terminal end. The overall FTIR data of compounds **6a–j** (Series 1) and **8a–j** (Series 2) are summarized in Table 1.

### 2.3. NMR Spectral Discussion

The ^1^H-NMR spectrum for intermediate **1** showed a small broad singlet at δ 10.70 ppm which was assigned to the hydroxyl proton while four doublets in the aromatic region (δ 6.90–8.40 ppm) were assigned to four different aromatic protons. Based on the ^1^H and ^13^C-NMR spectra of intermediates **2a–e**, signals in the upfield region were assigned for protons (δ 0.90–4.00 ppm) and carbons (δ 14.00–68.60 ppm) due to the successful of alkylation reaction. Intermediates **1** and **2a–e** were further reduced to form intermediates **3a–f**. A singlet at δ 5.90 ppm in the ^1^H-NMR spectrum of **3a–f** was assigned to the N-H of the amine group. Meanwhile, the chemical structures of **3g–i** were consistent with the NMR spectral data.

The ^1^H and ^13^C-NMR spectra of hexasubstituted cyclotriphosphazene benzoate, **4** showed the signal for a methoxy proton and carbon in the upfield region at δ 3.90 ppm and δ 52.20 ppm, respectively. The carbonyl carbon can be observed at δ 165.10 ppm and four sets of aromatic carbons at δ 120.60, 126.90, 131.10, and 152.90 ppm. Intermediate **5** was synthesized from the hydrolysis of **4** which transformed the methyl benzoate group into benzoic acid. No methoxy proton and carbon signals were observed in both ^1^H and ^13^C-NMR spectra, which indicated that intermediate **4** has been completely hydrolyzed. The proton signal for the carboxyl (COOH) was unable to be observed as this proton known as a labile proton, making the signal not noticeable in the spectrum. Meanwhile, both intermediates **4** and **5** showed a singlet in the ^31^P-NMR spectrum, which confirmed that all the phosphorus atoms in the cyclotriphosphazene ring have been substituted with the same side arms.

In addition, another hexasubstituted cyclotriphosphazene intermediates (**7**, **9**, and **10**) revealed the same characteristic of the protons and carbons in intermediate **5**. In the ^1^H-NMR spectrum, only intermediate **9** showed an additional peak for N-H group and methyl proton at δ 9.90 and 2.00 ppm, respectively. The deprotection of **9** to **10** displayed the appearance of a new peak at δ 4.90 ppm for the NH_2_ group, which indicated that all the starting materials were fully reduced. In addition, all the intermediates showed a singlet in the ^31^P-NMR spectrum.

In Series 1, compounds **6a–j** with amide and azo linking unit showed six different set for the aromatic protons. The chemical shift values displayed a slight difference in these compounds due to the effect of different terminal groups, bond angles, and electronegativity of the molecules [41]. Compounds **6a–e** revealed the same pattern in both ^1^H and ^13^C-NMR spectra but only differed in the number of the alkyl chain at the terminal side arm. Moreover, an additional peak for the hydroxyl and amino proton can be observed in the ^1^H-NMR spectra for compounds **6f** and **6j**, respectively. For further investigation on the splitting patterns and chemical shifts, compound **6e** was used to represent the structure confirmation in Series 1. The structure of compound **6e** with complete atomic numbering is shown in Figure 4.

The ^1^H-NMR spectrum of compound **6e** (Figure 5) showed six different sets of doublets at δ 7.84, 7.70, 7.60, 7.05, 7.02, and 6.71 ppm were assigned to the aromatic protons, H3, H12, H9, H2, H13, and H8, respectively. Since the amide group withdrew the electron density of the molecule and resulted in greater deshielding effect, hence H3 was more deshielded compared to H12. The same reasoning applied to H2 with H13 and H8 with H9. However, the signals for H2 and H9 are almost overlapped and observed as two-attached doublets. H13 experienced less deshielding effect since this proton was located far from the amide group. The amide proton cannot be observed as this proton could form a hydrogen bond with the neighbouring atom, causing the signal not noticeable in the spectrum. Meanwhile, a triplet at δ 4.06 ppm belonged to the oxymethylene protons (H15), while all the methylene protons (H16-H27) were observed in the region of δ 1.32–1.77 ppm. Signal of the methyl protons, H28, was observed as a triplet in the most upfield region at δ 0.86 ppm. 

Based on the ^13^C-NMR spectrum of compound **6e** (Figure 6a), there were 27 carbon signals consisting of seven quaternary, six aromatics, one oxymethylene, twelve methylene, and one methyl carbon on the side arms. 

The quaternary carbons can be distinguished using the DEPT-NMR experiment. In this research, DEPT-90 and DEPT-135 have been used to differentiate the types of each carbon. In DEPT-90, only methine carbon (CH) appeared on the spectrum, while in DEPT-135, the methylene carbon (CH_2_) appeared at the negative signal whereas the methine (CH) and methyl (CH_3_) carbon showed at the positive signals. Seven quaternary carbons at δ 166.56 (C5), 160.64 (C1), 152.11 (C14), 147.46 (C10), 144.21 (C11), 135.92 (C7), and 129.03 ppm (C4) disappeared in both DEPT-90 (Figure 6b) and DEPT-135 spectra (Figure 6c). 

C4 experienced the least deshielding effect since this carbon was attached to the less electronegative carbon atom, while C5 was assigned to the carbonyl carbon. The peaks at δ 131.58, 124.80, 123.77, 120.80, 115.56, and 114.35 ppm corresponded to aromatic carbons C3, C12, C9, C2, C8, and C12, respectively. On the other hand, oxymethylene carbon (C15) was located at δ 68.74 ppm, followed by methylene carbons (C16–C27) in the region of δ 22.34–31.64 ppm and lastly, the methyl carbon, C28 resonated at δ 14.06 ppm in the most upfield region.

In Series 2, six doublets with the integrating of 12 aromatic protons in ^1^H-NMR can be observed in compounds **8a–i**. The signal for the heptyl, nonyl, decyl, dodecyl, and tetradecyl chains in compounds **8a–e**, respectively, were appeared at the upfield region of ^1^H and ^13^C-NMR. However, the signal for hydroxyl and carboxyl protons are not displayed in ^1^H-NMR due to their property as a labile proton in compounds **8f** and **8g**, respectively. There was a slight difference in the chemical shift values for compounds in both series, which corresponding to the effect of the amide-azo (Series 1) and azo-azo (Series 2). Hence, compound **8j** attached with the small substituent (NH_2_) was chosen to represent the structure confirmation of Series 2 as demonstrated in Figure 7. 

The ^1^H-NMR spectrum of compound **8j** (Figure 8) showed six different signals. Five different sets of aromatic protons were observed at δ 8.03, 7.70, 7.68, 6.79, and 6.69 ppm, which assigned to H3 and H6, H10, H2, H7, and H11, respectively. The assignment of these protons was done by comparing the activation level of the terminal groups. NH_2_ group was known as an electron donating group and the most active in electron donor substituents. It can donate the electron onto the carbon and few resonance structures can be drawn. As a result, electron density in the nucleus increased and the proton adjacent to NH_2_ group will be shifted to the most upfield region. Thus, H11 experienced the less deshielding effect compared to the other aromatic protons. Similarly, the signal of H10 was shifted upfield as compared to that of H3, which was more deshielded than H2. Meanwhile, the aromatic protons, H3 and H6 were overlapped and appeared as a triplet since they shared the similar chemical environment. H7 experienced the more shielding effect compared to that of H6 as these protons located near to NH_2_ group. Furthermore, a broad singlet at δ 6.13 ppm belonged to the amine protons, H13.

The ^13^C-NMR spectrum of compound **8j** (Figure 9a) showed six different quaternary and aromatic carbons. The aromatic carbons observed at δ 130.51, 126.85, 125.74, 121.37, 117.12, and 113.95 ppm were assigned to C3, C6, C10, C2, C7, and C11, respectively. C1 was located at the most downfield region as this carbon was directly attached to the electronegative oxygen atom. The carbons located at δ 169.83 (C1), 169.05 (C12), 153.75 (C5), 153.40 (C8), 143.46 (C4), and 137.28 (C9), which disappeared in the DEPT-45 spectrum as shown in Figure 9b, were assigned to the quaternary carbons.

The ^31^P-NMR spectrum of HCCP appeared at δ 20.00 ppm (Figure 10a). This behaviour was due to the effect of the hexa-functionality of chlorine atom in the HCCP. This phosphorus atom experienced more deshielding effect compared to compounds **6a–j** and **8a–j** as the chlorine atom decreased the electron density of the molecule. 

When the amide-azo linking units in compound **6e** (Series 1) introduced into the HCCP system, a singlet was clearly observed at δ 7.80 ppm (Figure 10b) which indicated that the reaction between intermediate **5** with **3e** was a success. Meanwhile, the phosphorus signal at δ 11.22 ppm (Figure 10c) appeared as a singlet for compound **8j** having azo-azo linking units (Series 2) corresponded to all the phosphorus in the cyclotriphosphazene ring having the same substituents in the side arms. An azo group is an electron acceptor which decreased the electron density in the molecules and thus resulted in a greater downfield shift in the phosphorus atoms. As a result, compounds with two azo linking units (Series 2) were shifted more downfield compared to compounds attached with one azo linking unit (Series 1).

### 2.4. Determination of Liquid Crystal Behaviour Using POM

Polarized optical microscope (POM) is a technique that employs polarized light to detect liquid crystal mesophase(s) using an Olympus system mesophase model bx53 linksys32. The sample between two glasses is placed on the hot stage under the microscope. The phase changes of the sample can be observed in the heating and cooling cycles, which can be controlled and recorded. Compounds with liquid crystal behaviour are known as mesogenic, while non-mesogenic molecules did not exhibited any mesophase texture. 

In this study, only intermediates **2a–e** with nitro substituent and alkoxy terminal chains showed a smectic A (SmA) phase in the heating and cooling cycles (Figure 11). The compounds in Series 1 with amide and azo linking units that bore a different terminal alkoxy chains are found to be mesogenic with smectic phases. Compounds **6a**,**b** with heptyl and nonyl chains have the texture of focal conic fans of SmA in both cycles (Figure 12). Interestingly, the tendency for the formation of smectic C (SmC) phase increased with an increasing number of aliphatic chains. SmC phase is a tilted analogue of the SmA phase with liquid-like layers, tilted with respect to the layer at angles which vary from compound to compound. Observation under POM (Figure 13) showed that compounds **6c–e** exhibited the broken focal-conic fans of SmC phase in both cycles.

A similar trend is observed for compounds **8a–e** (Series 2) having two azo linking units and alkoxy chains at the periphery. These compounds exhibit liquid crystal phase of broken focal-conic fans texture of SmC, as shown in Figure 14. In this works, other intermediates and both series of final compounds having small substituents such as hydroxyl, carboxyl, chloro, nitro, and amino groups in the side arms are found to be non-mesogenic. Only the transition of crystal to isotropic phase was observed in the heating cycle. Similar phase transitions were also observed in the cooling cycle, but in the reversed order form.

### 2.5. Determination of Thermal Transitions Using DSC

Differential scanning calorimetry (DSC) experiment was conducted in order to confirm the mesophase transition observed under POM. The thermal enthalpy, *ΔH* (kJ/mol) of each phase transition of **2a–e, 6a–e**, and **8a–e** is calculated and summarized in Table 2. 

The DSC thermogram of intermediates **2a–e** with a nitro group demonstrates two endotherms in the heating and cooling cycles. The thermogram shows the transitions of crystal to SmA phase, while further heating caused SmA to melt into the isotropic phase. In the cooling cycle, the transitions of the isotropic-SmA-crystal phases were observed.

For all compounds of Series 1 having alkoxy chains, two peaks in the DSC thermogram are observed. Compounds **6a** and **6b** (Appendix A), with their respective DSC thermograms, showed the formation of SmA from crystal to isotropic phase during heating cycle. Similar pattern of transition can be observed upon cooling cycle. Interestingly, an increased number of alkyl chains in compounds **6c–e** (Appendix A) caused the molecules to align with their long axes tilted relative to the layers which resulted in the formation of SmC phase. The thermogram of compound **6d** with two endotherms in both cycles is shown in Figure 15. In the heating cycle, the clearing temperature was observed at 168.55 °C for **6c**, 154.37 °C for **6d**, and 137.68 °C for **6e**.

Moreover, the DSC thermograms of compounds **8a–e** (Series 2) displayed two peaks in the heating and cooling cycles, which corresponded to the transitions of crystal to SmC and the isotropic phases. For compound **8c** (Figure 16), a shoulder in the first curve of the thermogram in the heating cycle indicated the transition of crystal to crystal phase before it was transformed into the SmC phase. However, the formation of crystal to crystal phase did not appear in the cooling cycle as there no shoulder was observed. The melting points for **8a–e** were observed at 116.65, 118.42, 113.54, 109.51, and 117.69 °C, while the clearing temperatures at 171.13, 169.24, 162.78, 160.57, and 151.96 °C, respectively. The DSC thermogram of these compounds are provided in the Appendix A. All the synthesized compounds exhibited smectic phases, showing a decrease in the melting and clearing temperatures when the alkoxy chain length was increased [25].

### 2.6. Structure Properties Relationship

In general, the skeleton structure of synthesized compounds must satisfy a certain requirement in order for the materials to have liquid crystal properties. Aromatic ring cores connected directly or through a linking unit are very useful in providing rigidity to the molecule. The ring system affects the liquid crystal stability and other physical properties, allowing a linear configuration [42]. Linking units serve to impart various functions to the structures of mesomorphic materials. They provide linking bridges between aromatic rings or interrupt conjugation between rings. Moreover, substituents are chosen to cover a wide range of steric and electronic nature, which represent the conjugated interaction with the central linking group via the intervening benzene rings. Hence, the suitable rigid cores, linking units and terminal groups are important aspects in a molecular modification which gives the impact on the ordering abilities of mesogenic molecules [43]

Mesophases for thermotropic LC molecules occur in a certain temperature range. If the temperature is too high, the thermal motion will destroy the delicate ordering of the liquid crystal phase, pushing the material into becoming the isotropic phase. If the temperature is too low, most liquid crystal materials will form a conventional crystal [10,44]. In this study, only intermediates **2a–e** (rod-like molecules) having aliphatic chains and nitro terminal group shows the liquid crystal behaviour of smectic A phase. This phenomenon was due to the properties of nitro substituent as an electron withdrawing group, which suggested that the π-stacking is favoured. The repulsive interactions between adjacent aromatic π-systems is maximized and then induce the mesophase transition [45]. The presence of alkoxy chains in **2a–e** adds flexibility to the rigid core and stabilize the molecular interactions needed for the formation of liquid crystal mesophase. However, the SmA phase disappeared when intermediates **2a****–****e** underwent a reduction reaction to form intermediates **3a****–****e** with an amino substituent at the periphery. This behaviour was attributes to the high interaction of the electron donating group of NH_2_ substituent with adjacent aromatic ring, which reduce the ability of molecules to adopt any liquid crystal phase. 

Meanwhile, final compounds **6a****–e** and **8a****–e** shows the smectic phase behaviour because these compounds favour a lamellar packing [46]. The molecules in smectic phase form layers in which the long axes of the molecules tend to be orthogonal to the layer planes. The molecules are aligned, and their cores are closely packed in layers, but can move freely within a layer. The layers are liquid-like in nature. Movement of molecules from one layer to another occurs quite freely, and the layers themselves are quite free to slide and move over on another [41]. It is reported that compounds with longer terminal chains were found to have a dramatic effect on the mesophase formation. Compounds with long alkyl chain length exhibited enantiotropic mesophase which is thermodynamically stable, while compounds with shorter alkyl chains showed monotropic mesophase since this compound has the unstable behaviour [47]. This is due to higher Van der Waals interactions and the possibility of intertwining between the alkoxy chains [41,46] 

Compounds with azo linking unit preserve a linearity to the molecule which enhance the anisotropic polarizability. Due to the rigid rod-like structure of the azo unit, the compound can adopt and allows them to behave as liquid crystal mesogens in many materials [32]. The incorporation of azo and amide groups led to higher rigidity due to the presence of the partial double bond character of the C-N bond which resulted in higher transition and clearing temperatures of the mesogens [47]. As a result, compounds in the first series have a higher melting and clearing temperatures compared to compound in the second series with two azo linking units. The partial double bond in C-N bond reduces the coplanarity of the molecule and the broadening of the rigid core, which resulted in the molecules that favoured a lamellar arrangement in the smectic layer. The formation of smectic phase for compounds in Series 2 is due to the enhanced polarizability of the molecules as the alkoxy chain length at one end increases [48]. However, the ratio of the terminal to lateral attractions was also low at a certain stage, which leads to the thermal vibration break down on the orientation of the molecules and results in only a smectic C phase in compounds **8a****–e**.

However, all the other compounds with small substituents such as hydroxyl, carboxyl, chloro, nitro, and amino groups do not exhibit any mesophase behaviour. Kelker and Hatz (1980) reported that compounds with terminal groups, such as hydroxyl, carboxyl, and amino, do not always form liquid crystal phases [49]. Chlorine is a polar substituent possessing strong dipole moment, thus induce the ability to promote mesomorphic properties [50,51]. The increased dipole moment can enhance the stability of lattice and rise the melting temperatures [52]. Moreover, the electronegativity of the chlorine atom reduces the degree of molecular order, which influences the steric hindrance in the central core of the molecule, and this enhances a higher clearing temperature. The effect of electron-withdrawing of nitro group as the terminal moiety suggested that the π-stacking is favoured by the addition of electron-withdrawing groups, which minimize the repulsive interactions between adjacent aromatic π-systems. Nevertheless, compounds with chloro and nitro substituents in both series for this study did not shows any liquid crystal behaviour. These small substituents tend to form a resonance effect with the aromatic ring which resulted in the cancellation of dipole moments of the molecule itself [53]. As a result, the compounds have lower polarizability which deactivates the benzene ring and did not able to induce the molecular interactions needed for the formation of liquid crystal mesophase [54]. The POM observation of compounds **6a****–j** and **8a–j** is summarized in Table 3.

### 2.7. Determination of Fire Retardant Properties Using LOI Testing

The fire-retardant properties of the hexasubstituted cyclotriphosphazene compounds were verified by limiting oxygen index (LOI) test using polyester resin as matrix for moulding. Polyester resin was chosen due to good mechanical properties, fast curing, low cost, and more sensitive to elevated temperatures. The sample was prepared by blending 1 wt% of the final compound in polyester resin. Methyl ethyl ketone peroxide (MEKP) was used as a curing agent and about 1 wt% of MEKP was added to the mixture. This mixture was stirred until the sample became homogeneous and then poured into the moulds. The samples were cured for 5 h in an oven at 60 °C and left overnight at room temperature before they were burned by LOI testing. The LOI test was performed using an FTT oxygen index, according to BS 2782: Part 1: Method 141 and ISO 4589 with the dimension of 120 mm × 10 mm × 4 mm. The data obtained were expressed in a form of percentage (%) and the LOI results were calculated according to the equation given below:
LOI = *C_F_* + (k × d)(1)
where *C_F_* is the oxygen concentration of the final test, k is the factor obtained from the manual book Fire Testing Technology (ISO 4589), and d is the oxygen concentration increment.

The LOI data in Table 4 show that pure polyester resin has the LOI value of 22.53%. The LOI value of polyester resin was increased to 24.71% when incorporated with 1 wt% of HCCP. The results showed that HCCP with six chlorine atoms as a starting material have a good thermal stability and fire retardancy [41]. The impact of the modification of the HCCP with an organic intermediate gave a positive result on the LOI value. The LOI values for the alkylated compounds decreased as the aliphatic chain length increased. This was due to the lower percentages of phosphorus and nitrogen in the high molecular weight compounds since the LOI value was influenced by the cyclotriphosphazene and linking units only [29]. The phosphorus content is an important factor for compounds to have good fire-retardant properties [55]. Higher phosphorus percentage in the compounds has been considered as a requirement for efficient fire retardants [56,57]. As a result, compounds with heptyl terminal chains have higher LOI values compared to that of the compounds with nonyl, decyl, dodecyl, and tetradecyl terminal chains.

Besides, the compounds with nitro and chloro substituents showed higher LOI values compared to those of the other compounds. These groups are able to release the electrons via resonance to the corresponding P-N bonds. Thus, the P-N synergistic effect was enhanced, and they exhibited both the condensed and gas phase action modes [58]. Similar to compounds with hydroxyl, carboxyl, and amino substituents at the terminal end, the compounds also have good LOI values. The presence of small groups increased the adhesion properties of the resin, corresponded to the existence of the lone pairs of the electron in the terminal groups which formed the inter hydrogen bonds, merely attraction between partially positive hydrogen and partially negative oxygen or negative atoms [59]. The existence of the hydrogen bonding was reflected in increased thermal stability and fire retardancy of the compounds. This molecule was transformed into a cross-linked structure, which caused an increase in the heat resistance in the laminate structure. Thus, these compounds able to demonstrate the flexibility and adhesion resistance towards combustion. The higher the thermal stability of the compounds, the less combustible the target materials.

Furthermore, compounds **6a****–****j** with amide and azo linking units have higher LOI values compared to those of compounds **8a****–j** with two azo linking units. This phenomenon could be attributed to the electron withdrawing properties of the amide moiety. The amide group enhanced the adhesion properties of the polyester resin through the hydrogen bonding thus increased the fire-retardant properties [60]. In this work, all the samples were prepared using the minimum of 1 wt% of additive usage. This method is used to achieve the highest fire retardancy. The results in Table 4 indicated that all the final compounds in both series have high fire retardancy as they achieved LOI values above 24.71% even though only 1 wt% of additive usage. Their effect is to reduce the initiation of a fire by delaying the spread of flame and provide resistance to ignition. Compounds **6i** and **8i** with nitro substituent showed the highest LOI values.

## 3. Materials and Methods

### 3.1. Chemicals

All the chemicals and solvents were used as received without further purification and purchased from Merck (Darmstadt, Germany), Qrëc (Asia) (Selangor, Malaysia), Sigma–Aldrich (Steinheim, Germany), Acros Organics (Geel, Belgium), Fluka (Shanghai, China), and BDH (British Drug Houses) (Nichiryo, Japan).

### 3.2. Instruments

In this work, the FTIR spectrum of each compound was obtained using the PerkinElmer FTIR Microscope Spotlight 200 spectrometer (PerkinElmer, Waltham, MA, USA). All the samples are scanned with ATR sampling accessory at range of 600–4000 cm^−1^. NMR spectroscopy is used to determine the molecular structure of a compound for certain atomic nuclei such as ^1^H, ^13^C, and ^31^P. All the NMR spectra were generated using a Bruker 500 MHz Ultrashield™ spectrometer (Bruker, Coventry, UK). CHN elemental analysis is used to determine the percentage of carbon (C), hydrogen (H), and nitrogen (N) in a sample. This technique involves a combustion of a sample in an excess of oxygen. The analysis of all the synthesized samples were carried out using the CHN analyzer, model PerkinElmer II, 2400 (PerkinElmer, Waltham, MA, USA). In addition, POM is an instrument used to observe the phase textures of a sample (Linkam, London, UK). Samples sandwiched between two round glass slides were placed on the hot stage of the microscope. This microscope is adjusted to obtain a clear resolution and the phase transitions was observed in the heating and cooling cycles. DSC is a quantitative thermoanalytical technique. The DSC experiment was carried out to further confirm the mesophase transition observed under POM. The sample is heated with a scan rate of 10 °C/min and held at its isotropic temperature for 2 min so as to attain the thermal stability. The cooling run is performed with the same scan rate of 10 °C/min. All the DSC thermograms were obtained using Pyris 1 Differential Scanning Calorimeter, PerkinElmer (PerkinElmer, Waltham, MA, USA). Finally, determination of the fire-retardant properties of final compounds was done using LOI experiment (S.S. Instruments Pvt. Ltd., Delhi, India). LOI testing was conducted in a closed chamber system occupied with oxygen and nitrogen gas inlets in order to control the atmosphere of the chamber. The sample was set vertically inside the chamber and then ignited. The atmosphere was adjusted and controlled to determine the minimum amount of oxygen required to burn a sample at a certain time. The minimum value of oxygen was calculated and expressed in the form of percentage.

### 3.3. Syntheses

#### 3.3.1. Synthesis of 4-((4-nitrophenyla)diazenyl)phenol, **1**

A solution of 4-nitroaniline (6.91 g, 0.05 mol) in 50 mL methanol was added to 13 mL of 12 M hydrochloric acid, HCl and the mixture was cooled at 0 °C. A solution of sodium nitrite, NaNO_2_ (3.45 g, 0.05 mol) in 15 mL of water was added dropwise to the mixture, which was left to stir for 30 min. The addition of phenol (4.71 g, 0.05 mol) in 70 mL of 20% sodium acetate hydrate, CH_3_COONa.3H_2_O to the mixture led to the formation of an orange precipitate. About 100 mL of water was then added to the solution and the reaction was left for 2 h at 0 °C and another 2 h at room temperature. The reaction progress was monitored by TLC. Upon completion, the precipitate was filtered, washed with water and dried overnight in open air. The product was recrystallised from methanol. Yield: 11.56 g (95.14%), mp: 199.2–203.5 °C, orange powder. FTIR (cm^−1^): 3250 (O-H stretching), 1504 (aromatic C=C stretching), 1458 (N=N stretching), 1326 (C-O stretching), 1173 (C-N stretching). ^1^H-NMR (500 MHz, DMSO-d_6_) δ, ppm: 10.70 (s, 1H), 8.38 (d, *J* = 10.0 Hz, 2H), 7.97 (d, *J* = 10.0 Hz, 2H), 7.87 (d, *J* = 5.0 Hz, 2H), 6.97 (d, *J* = 10.0 Hz, 2H). ^13^C-NMR (125 MHz, DMSO-d_6_) δ, ppm: 162.31, 155.54, 147.76, 145.39, 125.77, 125.00, 122.96, 116.21. CHN elemental analysis: Calculated for C_12_H_9_N_3_O_3_: C: 59.26%, H: 3.73%, N: 17.28%; Found: C: 58.69%, H: 3.71%, N: 16.98%.

#### 3.3.2. Synthesis of (4-alkyloxyphenyl)-(4-nitrophenylazo)diazene, **2a–e**

(4-Heptyloxyphenyl)-(4-nitrophenylazo)diazene, **2a**: A solution of intermediate **1** (10.00 g, 0.04 mol) in 20 mL DMF and 1-bromoheptane (7.16 g, 0.04 mol) in 20 mL DMF were mixed in a 100 mL round bottom flask. Potassium carbonate (11.06 g, 0.08 mol) and potassium iodide (0.66 g, 4.00 mmol) were added to the mixture, which was refluxed for 12 h. The reaction was monitored using TLC. Upon completion, the mixture was poured into 500 mL cold water and orange-red precipitate began to form. The product was filtered, washed with water and dried in open air. The same method was used to synthesize **2b****–e**. Yield: 11.32 g (82.33%), mp: 97.4–98.8 °C, orange-red powder. FTIR (cm^−1^): 2927 and 2857 (C*sp^3^*-H stretching), 1601 (aromatic C=C stretching), 1468 (N=N stretching), 1246 (C-O stretching), 1141 (C-N stretching). ^1^H-NMR (500 MHz, DMSO-d_6_) δ, ppm: 8.34 (d, *J* = 10.0 Hz, 2H), 7.96 (d, *J* = 10.0 Hz, 2H), 7.94 (d, *J* = 10.0 Hz, 2H), 7.01 (d, *J* = 5.0 Hz, 2H), 4.04 (t, *J* = 7.5 Hz, 2H), 1.79–1.83 (m, 2H), 1.43–1.48 (m, 2H), 1.28–1.38 (m, 6H), 0.89 (t, *J* = 7.5 Hz, 3H). ^13^C-NMR (125 MHz, DMSO-d_6_) δ, ppm: 162.98, 156.11, 148.23, 146.82, 125.62, 124.70, 123.08, 114.95, 68.56, 31.76, 29.15, 29.03, 25.96, 22.60, 14.06. CHN elemental analysis: Calculated for C_19_H_23_N_3_O_3_: C: 66.84%, H: 6.79%, N: 12.31%; Found: C: 67.18%, H: 6.77%, N: 12.19%.

(4-Nonyloxyphenyl)-(4-nitrophenylazo)diazene, **2b**: Yield: 12.11 g (82.05%), mp: 96.7–98.1 °C, light red powder. FTIR (cm^−1^): 2921 and 2851 (C*sp^3^*-H stretching), 1601 (aromatic C=C stretching), 1466 (N=N stretching), 1248 (C-O stretching), 1141 (C-N stretching). ^1^H-NMR (500 MHz, DMSO-d_6_) δ, ppm: 8.34 (d, *J* = 10.0 Hz, 2H), 7.96 (d, *J* = 10.0 Hz, 2H), 7.94 (d, *J* = 10.0 Hz, 2H), 7.01 (d, *J* = 5.0 Hz, 2H), 4.05 (t, *J* = 7.5 Hz, 2H), 1.78–1.84 (m, 2H), 1.43–1.49 (m, 2H), 1.26–1.37 (m, 10H), 0.87 (t, *J* = 5.0 Hz, 3H). ^13^C-NMR (125 MHz, DMSO-d_6_) δ, ppm: 162.99, 156.08, 148.21, 146.80, 125.63, 124.69, 123.08, 114.94, 68.56, 31.88, 29.53, 29.38, 29.26, 29.15, 26.00, 22.68, 14.11. CHN elemental analysis: Calculated for C_21_H_27_N_3_O_3_: C: 68.27%, H: 7.37%, N: 11.37%; Found: C: 67.98%, H: 7.31%, N: 11.28%.

(4-Decyloxyphenyl)-(4-nitrophenylazo)diazene, **2c**: Yield: 13.89 g (90.67%), mp: 95.3–97.9 °C, red powder. FTIR (cm^−1^): 2921 and 2851 (C*sp^3^*-H stretching), 1601 (aromatic C=C stretching), 1468 (N=N stretching), 1248 (C-O stretching), 1141 (C-N stretching). ^1^H-NMR (500 MHz, DMSO-d_6_) δ, ppm: 8.34 (d, *J* = 5.0 Hz, 2H), 7.96 (d, *J* = 5.0 Hz, 2H), 7.94 (d, *J* = 10.0 Hz, 2H), 7.00 (d, *J* = 10.0 Hz, 2H), 4.04 (t, *J* = 5.0 Hz, 2H), 1.78–1.84 (m, 2H), 1.43–1.48 (m, 2H), 1.26–1.37 (m, 12H), 0.87 (t, *J* = 7.5 Hz, 3H). ^13^C-NMR (125 MHz, DMSO-d_6_) δ, ppm: 162.99, 156.08, 148.20, 146.80, 125.62, 124.66, 123.07, 114.93, 68.56, 31.90, 29.57, 29.56, 29.38, 29.32, 29.15, 26.00, 22.68, 14.10. CHN elemental analysis: Calculated for C_22_H_29_N_3_O_3_: C: 68.90%, H: 7.62%, N: 10.96%; Found: C: 68.80%, H: 7.53%, N: 10.88%.

(4-Dodecyloxyphenyl)-(4-nitrophenylazo)diazene, **2d**: Yield: 14.98 g (91.11%), mp: 95.1–97.3 °C, red powder. FTIR (cm^−1^): 2916 and 2849 (C*sp^3^*-H stretching), 1601 (aromatic C=C stretching), 1468 (N=N stretching), 1248 (C-O stretching), 1143 (C-N stretching). ^1^H-NMR (500 MHz, DMSO-d_6_) δ, ppm: 8.34 (d, *J* = 10.0 Hz, 2H), 7.96 (d, *J* = 10.0 Hz, 2H), 7.94 (d, *J* = 10.0 Hz, 2H), 7.00 (d, *J* = 5.0 Hz, 2H), 4.04 (t, *J* = 7.5 Hz, 2H), 1.78–1.84 (m, 2H), 1.43–1.48 (m, 2H), 1.22–1.37 (m, 16H), 0.86 (t, *J* = 7.5 Hz, 3H). ^13^C-NMR (125 MHz, DMSO-d_6_) δ, ppm: 162.99, 156.08, 148.21, 146.80, 125.62, 124.67, 123.07, 114.93, 68.56, 31.93, 29.67, 29.64, 29.60, 29.57, 29.38, 29.36, 29.15, 26.00, 22.69, 14.11. CHN elemental analysis: Calculated for C_24_H_33_N_3_O_3_: C: 70.04%, H: 8.08%, N: 10.21%; Found: C: 69.87%, H: 7.99%, N: 10.14%.

(4-Tetradecyloxyphenyl)-(4-nitrophenylazo)diazene, **2e**: Yield: 15.44 g (87.93%), mp: 94.3–95.9 °C, dark-red powder. FTIR (cm^−1^): 2913 and 2849 (C*sp^3^*-H stretching), 1603 (aromatic C=C stretching), 1468 (N=N stretching), 1248 (C-O stretching), 1143 (C-N stretching). ^1^H-NMR (500 MHz, DMSO-d_6_) δ, ppm: 8.34 (d, *J* = 10.0 Hz, 2H), 7.96 (d, *J* = 10.0 Hz, 2H), 7.94 (d, *J* = 10.0 Hz, 2H), 7.01 (d, *J* = 5.0 Hz, 2H), 4.04 (t, *J* = 7.5 Hz, 2H), 1.78–1.84 (m, 2H), 1.43–1.48 (m, 2H), 1.22–1.37 (m, 20H), 0.86 (t, *J* = 5.0 Hz, 3H). ^13^C-NMR (125 MHz, DMSO-d_6_) δ, ppm: 162.98, 156.11, 148.24, 146.82, 125.62, 124.70, 123.08, 114.95, 68.56, 31.92, 29.69, 29.67, 29.65, 29.58, 29.55, 29.35, 29.14, 25.99, 22.68, 14.10, two carbon environments not observed due to coincident resonances. CHN elemental analysis: Calculated for C_26_H_37_N_3_O_3_: C: 71.04%, H: 8.48%, N: 9.56%; Found: C: 71.17%, H: 8.40%, N: 9.45%.

#### 3.3.3. Synthesis of 4-((4-substitutedphenyl)diazenyl)aniline, **3a–f**

4-((4-Heptyloxyphenyl)diazenyl)aniline, **3a**: In a 250 mL round bottom flask, a solution of sodium sulphide hydrate, Na_2_S.9H_2_O (6.24 g, 0.08 mol) in 20 mL ethanol and 20 mL water was mixed with a solution of intermediate **2a** (6.82 g, 0.02 mmol) in 40 mL ethanol. The mixture was stirred and refluxed for 12 h. The reaction progress was monitored using TLC. Upon completion, the mixture was cooled in an ice bath. The precipitate formed was filtered, washed with cold ethanol and dried in open air. The same method was used to synthesize **3b****–f**. Yield: 5.35 g (86.01%), mp: 130.3–132.4 °C, orange powder. FTIR (cm^−1^): 3470 and 3338 (N-H stretching), 2932 and 2851 (C*sp^3^*-H stretching), 1598 (aromatic C=C stretching), 1498 (N=N stretching), 1248 (C-O stretching), 1141 (C-N stretching). ^1^H-NMR (500 MHz, DMSO-d_6_) δ, ppm: 7.71 (d, *J* = 10.0 Hz, 2H), 7.61 (d, *J* = 10.0 Hz, 2H), 7.02 (d, *J* = 10.0 Hz, 2H), 6.66 (d, *J* = 10.0 Hz, 2H), 5.93 (s, 2H), 4.00 (t, *J* = 7.5 Hz, 2H), 1.68–1.74 (m, 2H), 1.36–1.42 (m, 2H), 1.22–1.34 (m, 6H), 0.85 (t, *J* = 7.5 Hz, 3H). ^13^C-NMR (125 MHz, DMSO-d_6_) δ, ppm: 159.89, 152.04, 146.40, 142.84, 124.54, 123.35, 114.80, 113.43, 67.80, 31.16, 28.59, 28.40, 21.98, 13.88, one carbon environments not observed due to coincident resonances. CHN elemental analysis: Calculated for C_19_H_25_N_3_O: C: 73.28%, H: 8.09%, N: 13.49%; Found: C: 73.11%, H: 8.07%, N: 13.37%.

4-((4-Nonyloxyphenyl)diazenyl)aniline, **3b**: Yield: 5.80 g (85.55%), mp: 128.7–130.5 °C, dark orange powder. FTIR (cm^−1^): 3465 and 3333 (N-H stretching), 2916 and 2849 (C*sp^3^*-H stretching), 1598 (aromatic C=C stretching), 1498 (N=N stretching), 1243 (C-O stretching), 1143 (C-N stretching). ^1^H-NMR (500 MHz, DMSO-d_6_) δ, ppm: 7.71 (d, *J* = 10.0 Hz, 2H), 7.61 (d, *J* = 5.0 Hz, 2H), 7.03 (d, *J* = 10.0 Hz, 2H), 6.66 (d, *J* = 5.0 Hz, 2H), 5.94 (s, 2H), 4.01 (t, *J* = 7.5 Hz, 2H), 1.69–1.74 (m, 2H), 1.38–1.44 (m, 2H), 1.22–1.35 (m, 10H), 0.85 (t, *J* = 5.0 Hz, 3H). ^13^C-NMR (125 MHz, DMSO-d_6_) δ, ppm: 159.88, 152.08, 146.43, 142.84, 124.54, 123.33, 114.77, 113.39, 67.79, 31.23, 28.91, 28.72, 28.60, 28.44, 22.04, 13.90, one carbon environments not observed due to coincident resonances. CHN elemental analysis: Calculated for C_21_H_29_N_3_O: C: 74.30%, H: 8.61%, N: 12.38%; Found: C: 73.91%, H: 8.55%, N: 12.33%.

4-((4-Decyloxyphenyl)diazenyl)aniline, **3c**: Yield: 6.23 g (81.33%), mp: 127.9–130.1 °C, orange powder. FTIR (cm^−1^): 3478 and 3352 (N-H stretching), 2919 and 2849 (C*sp^3^*-H stretching), 1595 (aromatic C=C stretching), 1498 (N=N stretching), 1248 (C-O stretching), 1146 (C-N stretching). ^1^H-NMR (500 MHz, DMSO-d_6_) δ, ppm: 7.70 (d, *J* = 10.0 Hz, 2H), 7.60 (d, *J* = 5.0 Hz, 2H), 7.01 (d, *J* = 10.0 Hz, 2H), 6.66 (d, *J* = 5.0 Hz, 2H), 5.91 (s, 2H), 3.98 (t, *J* = 7.5 Hz, 2H), 1.66–1.72 (m, 2H), 1.35–1.41 (m, 2H), 1.19–1.30 (m, 12H), 0.82 (t, *J* = 5.0 Hz, 3H). ^13^C-NMR (125 MHz, DMSO-d_6_) δ, ppm: 159.87, 152.01, 146.37, 142.83, 124.53, 123.35, 114.75, 113.43, 67.76, 31.22, 28.91, 28.86, 28.67, 28.61, 28.55, 25.40, 22.03, 13.88. CHN elemental analysis: Calculated for C_22_H_31_N_3_O: C: 74.75%, H: 8.84%, N: 11.89%; Found: C: 74.66%, H: 8.78%, N: 11.77%.

4-((4-Dodecyloxyphenyl)diazenyl)aniline, **3d**: Yield: 6.90 g (90.55%), mp: 125.7–128.3 °C, dark-red powder. FTIR (cm^−1^): 3473 and 3333 (N-H stretching), 2919 and 2851 (C*sp^3^*-H stretching), 1601 (aromatic C=C stretching), 1498 (N=N stretching), 1248 (C-O stretching), 1149 (C-N stretching). ^1^H-NMR (500 MHz, DMSO-d_6_) δ, ppm: 7.66 (d, *J* = 10.0 Hz, 2H), 7.58 (d, *J* = 10.0 Hz, 2H), 6.89 (d, *J* = 10.0 Hz, 2H), 6.64 (d, *J* = 5.0 Hz, 2H), 5.86 (s, 2H), 3.84 (t, *J* = 7.5 Hz, 2H), 1.57–1.63 (m, 2H), 1.26–1.32 (m, 2H), 1.10–1.22 (m, 16H), 0.77 (t, *J* = 7.5 Hz, 3H). ^13^C-NMR (125 MHz, DMSO-d_6_) δ, ppm: 159.80, 151.93, 146.39, 142.89, 124.49, 123.27, 114.48, 113.37, 67.64, 31.30, 29.08, 29.05, 29.03, 29.01, 28.80, 28.75, 28.63, 25.46, 22.07, 13.76. CHN elemental analysis: Calculated for C_24_H_35_N_3_O: C: 75.55%, H: 9.25%, N: 11.01%; Found: C: 75.43%, H: 9.19%, N: 10.91%.

4-((4-Tetradecyloxyphenyl)diazenyl)aniline, **3e**: Yield: 7.15 g (87.41%), mp: 124.1–126.7 °C, red-brown powder. FTIR (cm^−1^): 3462 and 3328 (N-H stretching), 2919 and 2849 (C*sp^3^*-H stretching), 1601 (aromatic C=C stretching), 1498 (N=N stretching), 1251 (C-O stretching), 1146 (C-N stretching). ^1^H-NMR (500 MHz, DMSO-d_6_) δ, ppm: 7.71 (d, *J* = 5.0 Hz, 2H), 7.60 (d, *J* = 10.0 Hz, 2H), 7.03 (d, *J* = 5.0 Hz, 2H), 6.65 (d, *J* = 10.0 Hz, 2H), 5.93 (s, 2H), 4.01 (t, *J* = 7.5 Hz, 2H), 1.69–1.74 (m, 2H), 1.37–1.43 (m, 2H), 1.20–1.33 (m, 20H), 0.84 (t, *J* = 7.5 Hz, 3H). ^13^C-NMR (125 MHz, DMSO-d_6_) δ, ppm: 159.87, 152.06, 146.41, 142.83, 124.53, 123.33, 114.77, 113.40, 67.77, 31.23, 28.99, 28.98, 28.96, 28.95, 28.94, 28.90, 28.88, 28.64, 28.56, 25.39, 22.03, 13.89. CHN elemental analysis: Calculated for C_26_H_39_N_3_O: C: 76.24%, H: 9.60%, N: 10.26%; Found: C: 75.55%, H: 9.57%, N: 10.22%.

4-((4-Aminophenyl)diazenyl)phenol, **3f**: Intermediate **1** (9.72 g, 0.04 mol) was used in the reaction. Yield: 7.00 g (81.78%), mp: 208.5–211.1 °C, orange-brown powder. FTIR (cm^−1^): 3352 and 3298 (NH_2_ stretching), 3210 (O-H stretching), 1609 (aromatic C=C stretching), 1490 (N=N stretching), 1256 (C-O stretching), 1176 (C-N stretching). ^1^H-NMR (500 MHz, DMSO-d_6_) δ, ppm: 7.64 (d, *J* = 5.0 Hz, 2H), 7.58 (d, *J* = 10.0 Hz, 2H), 6.87 (d, *J* = 10.0 Hz, 2H), 6.66 (d, *J* = 10.0 Hz, 2H), 5.82 (s, 2H), one exchangeable proton was not observed. ^13^C-NMR (125 MHz, DMSO-d_6_) δ, ppm: 159.81, 152.14, 145.89, 143.42, 124.82, 124.11, 116.26, 114.01. CHN elemental analysis: Calculated for C_12_H_11_N_3_O: C: 67.59%, H: 5.20%, N: 19.71%; Found: C: 67.11%, H: 5.22%, N: 19.55%.

#### 3.3.4. Synthesis of 4-((4-aminophenyl)diazenyl)benzoic acid, **3g**

A mixture of 1,4-phenylenediamine (19.44 g, 0.18 mol) and 4-nitrobenzoic acid (16.70 g, 0.1 mol) was placed in 500 mL round bottom flask containing 200 mL of 3% aqueous NaOH. The reaction mixture was stirred and refluxed for 12 h. The reaction progress was monitored by TLC. Upon completion, the mixture was cooled into an ice bath. The precipitate formed was filtered, washed a few times with cold water and dried overnight in open air to give a brown solid. Yield: 17.65 g (73.24%), mp: 220.7–222.4 °C, brown powder. FTIR (cm^−1^): 3211 (O-H stretching), 3336 and 3207 (N-H stretching), 1700 (C=O stretching), 1601 (aromatic C=C stretching), 1498 (N=N stretching), 1238 (C-O stretching), 1143 (C-N stretching). ^1^H-NMR (500 MHz, DMSO-d_6_) δ, ppm: 7.96 (d, *J* = 5.0 Hz, 2H), 7.66 (d, *J* = 10.0 Hz, 2H), 7.64 (d, *J* = 5.0 Hz, 2H), 6.68 (d, *J* = 10.0 Hz, 2H), 6.03 (s, 2H), one exchangeable proton was not observed. ^13^C-NMR (125 MHz, DMSO-d_6_) δ, ppm: 169.88, 153.19, 153.15, 143.47, 141.25, 130.38, 125.58, 121.15, 113.97. CHN elemental analysis: Calculated for C_13_H_11_N_3_O_2_: C: 64.72%, H: 4.60%, N: 17.42%; Found: C: 64.33%, H: 4.61%, N: 17.37%.

#### 3.3.5. Synthesis of 4-((4-substitutedphenyl)diazenyl)aniline, **3h–i**

4-((4-Chlorophenyl)diazenyl)aniline, **3h**: Part 1: A mixture of aniline (4.65 g, 0.05 mol), sodium bisulphite (10.41 g, 0.1 mol), and 10 mL of aqueous formaldehyde were placed in 100 mL round bottom flask and then stirred at room temperature for 2 h. Then, 50 mL of water was added to form a clear solution. Part 2: 4-Chloroaniline (6.42 g, 0.05 mol) was dissolved in 50 mL of methanol, followed by the addition of 13 mL of 12 M hydrochloric acid, HCl. The mixture was stirred and cooled to 0 °C. A solution of sodium nitrite, NaNO_2_ (3.45 g, 0.05 mol) in 15 mL of water was added dropwise to the mixture and was left for 30 min. Part 3: Sodium bicarbonate (8.40 g) was added to the solution prepared in Part 1. The mixture was cooled at 0 °C and was then poured slowly into the solution prepared in Part 2. The mixture was stirred for 12 h at room temperature and the precipitate formed was filtered. The precipitate was then added in 20 mL of 10% NaOH solution and was refluxed for 2 h. All the reaction progress was monitored by TLC. Upon completion, the precipitate was filtered and dried overnight in open air to form a dark orange solid. The same method was used for synthesis of **3i**. Yield: 8.50 g (73.13%), mp: 175.7–178.1 °C, dark orange powder. FTIR (cm^−1^): 3336 and 3207 (N-H stretching), 1595 (aromatic C=C stretching), 1496 (N=N stretching), 1141 (C-N stretching), 785 (C-Cl bending). ^1^H-NMR (500 MHz, DMSO-d_6_) δ, ppm: 7.62 (d, *J* = 10.0 Hz, 2H), 7.56 (d, *J* = 10.0 Hz, 2H), 6.86 (d, *J* = 10.0 Hz, 2H), 6.66 (d, *J* = 5.0 Hz, 2H), 5.69 (s, 2H). ^13^C-NMR (125 MHz, DMSO-d_6_) δ, ppm: 159.36, 152.00, 145.97, 143.46, 124.85, 124.13, 116.26, 114.21. CHN elemental analysis: Calculated for C_12_H_10_N_3_: C: 62.21%, H: 4.35%, N: 18.14%; Found: C: 62.07%, H: 4.33%, N: 17.98%.

4-((4-Nitrophenyl)diazenyl)aniline, **3i**: Yield: 9.00 g (74.07%), mp: 180.1–182.7 °C, brown powder. FTIR (cm^−1^): 3352 and 3209 (N-H stretching), 1600 (aromatic C=C stretching), 1498 (N=N stretching), 1143 (C-N stretching). ^1^H-NMR (500 MHz, DMSO-d_6_) δ, ppm: 8.34 (d, *J* = 10.0 Hz, 2H), 7.89 (d, *J* = 10.0 Hz, 2H), 7.73 (d, *J* = 10.0 Hz, 2H), 6.70 (d, *J* = 5.0 Hz, 2H), 6.49 (s, 2H). ^13^C-NMR (125 MHz, DMSO-d_6_) δ, ppm: 162.77, 156.01, 148.22, 145.85, 126.24, 125.46, 123.43, 116.67. CHN elemental analysis: Calculated for C_12_H_10_N_4_O_2_: C: 59.50%, H: 4.16%, N: 23.13%; Found: C: 58.91%, H: 4.14%, N: 22.99%.

#### 3.3.6. Synthesis of Hexakis(4-benzoate-phenoxy)cyclotriphosphazene, **4**

In a 250 mL round bottom flask, a mixture of methyl 4-hydroxybenzoate (10.64 g, 0.07 mol), phosphonitrilic chloride trimer (3.48 g, 0.01 mol), and potassium carbonate, K_2_CO_3_ (13.82 g, 0.1 mol) were placed in 150 mL acetone and then refluxed for 4 days. The reaction progress was monitored using TLC. Upon completion, the mixture was poured into 250 mL of cool water. The precipitate formed was filtered, washed with water and dried overnight to give a white solid. Yield: 9.30 g (89.34%), mp: 208.3–210.8 °C, white powder. FTIR (cm^−1^): 2875 and 2930 (C*sp^3^*-H stretching), 1708 (C=O stretching), 1606 (aromatic C=C stretching), 1253 (C-O stretching), 1189 (P=N stretching), 1172 (C-N stretching), 950 (P-O-C stretching). ^1^H-NMR (500 MHz, DMSO-d_6_) δ, ppm: 7.78 (d, *J* = 10.0 Hz, 2H), 7.05 (d, *J* = 5.0 Hz, 2H), 3.87 (s, 3H). ^13^C-NMR (125 MHz, DMSO-d_6_) δ, ppm: 165.13, 152.85, 131.08, 126.91, 120.62, 52.23. ^31^P-NMR (500 MHz, DMSO-d_6_) δ, ppm: 8.00 (s, 1P). CHN elemental analysis: Calculated for C_48_H_42_N_3_O_18_P_3_: C: 55.34%, H: 4.06%, N: 4.03%; Found: C: 55.11%, H: 4.05%, N: 3.99%.

#### 3.3.7. Synthesis of Hexakis(4-carboxy-phenoxy)cyclotriphosphazene, **5**

Intermediate **4** (8.50 g, 8.17 mmol) and sodium hydroxide, NaOH (5.88 g, 0.12 mol) in 150 mL of ethanol were mixed in a 250 mL round bottom flask. The mixture was refluxed for 5 h. White solution was formed and the reaction progress was monitored by TLC. Upon completion, the mixture was poured into 250 mL cooled water. A clear solution was observed, which then acidified with HCl until the precipitate was formed. The precipitate was filtered, washed with water and dried to give a white solid. Yield: 6.90 g (88.30%), mp: 217.1–219.5 °C, white powder. FTIR (cm^−1^): 3250 (O-H stretching), 1695 (C=O stretching), 1601 (aromatic C=C stretching), 1211 (C-O stretching), 1184 (P=N stretching), 1157 (C-N stretching), 951 (P-O-C stretching). ^1^H-NMR (500 MHz, DMSO-d_6_) δ, ppm: 7.83 (d, *J* = 10.0 Hz, 2H), 6.99 (d, *J* = 10.0 Hz, 2H), one exchangeable proton was not observed. ^13^C-NMR (125 MHz, DMSO-d_6_) δ, ppm: 166.25, 152.72, 131.22, 128.24, 120.49. ^31^P-NMR (500 MHz, DMSO-d_6_) δ, ppm: 8.04 (s, 1P). CHN elemental analysis: Calculated for C_42_H_30_N_3_O_18_P_3_: C: 52.68%, H: 3.16%, N: 4.39%; Found: C: 52.47%, H: 3.17%, N: 4.37%.

#### 3.3.8. Synthesis of Hexakis{4-((E)-(4-((E)-4-substituted-phenyl)diazenyl)phenyl)benzamide} triazaphosphazene, **6a–i**

Hexakis{4-((*E*)-(4-((*E*)-4-heptyloxy-phenyl)diazenyl)phenyl)benzamide}triazaphosphazene, **6a**: A mixture of intermediate **5** (1.00 g, 1.05 mmol), thionyl chloride (0.87 g, 7.32 mmol) and dichloromethane, DCM (20 mL) was mixed in a 100 mL round bottom flask. The mixture was stirred at room temperature for 6 h and an acid chloride was formed (in-situ reaction) by replacing the carboxyl function of intermediate **5**. Without isolation, acid chloride was further reacted with intermediate **3a** (2.11 g, 6.79 mmol) in 10 mL tetrahydrofuran, THF. Triethylamine, Et_3_N (0.32 g, 3.14 mmol) was added dropwise to the mixture which was stirred at room temperature for 4 days. The reaction progress was monitored by TLC. Upon completion, the product was filtered, and the filtrate was evaporated until the precipitate was formed. This precipitate was recrystallised from ethanol to give a yellow solid. The same method was used to synthesize **6b****–i**. Yield: 2.00 g (70.50%), mp: 155.3–157.7 °C, yellow-orange powder. FTIR (cm^−1^): 3342 (N-H stretching), 2924 and 2860 (C*sp^3^*-H stretching), 1703 (C=O stretching), 1591 (aromatic C=C stretching), 1485 (N=N stretching), 1248 (C-O stretching), 1186 (P=N stretching), 1149 (C-N stretching), 1017 (P-O-C stretching). ^1^H-NMR (500 MHz, DMSO-d_6_) δ, ppm: 7.82 (d, *J* = 10.0 Hz, 2H), 7.69 (d, *J* = 10.0 Hz, 2H), 7.58 (d, *J* = 10.0 Hz, 2H), 7.03 (d, *J* = 5.0 Hz, 2H), 7.01 (d, *J* = 5.0 Hz, 2H), 6.73 (d, *J* = 5.0 Hz, 2H), 4.04 (t, *J* = 7.5 Hz, 2H), 1.70–1.75 (m, 2H), 1.39–1.45 (m, 2H), 1.21–1.35 (m, 6H), 0.85 (t, *J* = 7.5 Hz, 3H), one exchangeable proton was not observed. ^13^C-NMR (125 MHz, DMSO-d_6_) δ, ppm: 166.67, 160.69, 151.73, 147.38, 144.36, 137.03, 131.58, 128.91, 124.77, 123.82, 120.80, 115.63, 114.61, 68.79, 31.55, 29.18, 28.83, 25.82, 22.28, 14.02. ^31^P-NMR (500 MHz, DMSO-d_6_) δ, ppm: 7.81 (s, 1P). CHN elemental analysis: Calculated for C_156_H_168_N_21_O_18_P_3_: C: 68.93%, H: 6.23%, N: 10.82%; Found: C: 68.77%, H: 6.20%, N: 10.75%.

Hexakis{4-((*E*)-(4-((*E*)-4-nonyloxy-phenyl)diazenyl)phenyl)benzamide}triazaphosphazene, **6b**: Yield: 1.93 g (64.07%), mp: 147.6–149.4 °C, orange powder. FTIR (cm^−1^): 3343 (N-H stretching), 2921 and 2851 (C*sp^3^*-H stretching), 1695 (C=O stretching), 1593 (aromatic C=C stretching), 1483 (N=N stretching), 1211 (C-O stretching), 1186 (P=N stretching), 1160 (C-N stretching), 1017 (P-O-C stretching). ^1^H-NMR (500 MHz, DMSO-d_6_) δ, ppm: 7.82 (d, *J* = 10.0 Hz, 2H), 7.69 (d, *J* = 10.0 Hz, 2H), 7.59 (d, *J* = 10.0 Hz, 2H), 7.03 (d, *J* = 5.0 Hz, 2H), 7.01 (d, *J* = 5.0 Hz, 2H), 6.73 (d, *J* = 5.0 Hz, 2H), 4.04 (t, *J* = 7.5 Hz, 2H), 1.70–1.75 (m, 2H), 1.39–1.45 (m, 2H), 1.22–1.35 (m, 10H), 0.85 (t, *J* = 7.5 Hz, 3H), one exchangeable proton was not observed. ^13^C-NMR (125 MHz, DMSO-d_6_) δ, ppm: 166.67, 160.69, 151.73, 147.38, 144.36, 137.03, 131.58, 128.91, 124.77, 123.82, 120.80, 115.63, 114.61, 68.79, 31.55, 29.18, 29.08, 29.02, 28.83, 25.82, 22.28, 14.02. ^31^P-NMR (500 MHz, DMSO-d_6_) δ, ppm: 7.82 (s, 1P). CHN elemental analysis: Calculated for C_168_H_192_N_21_O_18_P_3_: C: 69.91%, H: 6.71%, N: 10.19%; Found: C: 69.82%, H: 6.73%, N: 10.11%.

Hexakis{4-((*E*)-(4-((*E*)-4-decyloxy-phenyl)diazenyl)phenyl)benzamide}triazaphosphazene, **6c**: Yield: 2.12 g (68.38%), mp: 144.1–146.6 °C, yellow powder. FTIR (cm^−1^): 3343 (N-H stretching), 2919 and 2849 (C*sp^3^*-H stretching), 1702 (C=O stretching), 1593 (aromatic C=C stretching), 1488 (N=N stretching), 1216 (C-O stretching), 1184 (P=N stretching), 1149 (C-N stretching), 1019 (P-O-C stretching). ^1^H-NMR (500 MHz, DMSO-d_6_) δ, ppm: 7.85 (d, *J* = 5.0 Hz, 2H), 7.70 (d, *J* = 10.0 Hz, 2H), 7.60 (d, *J* = 10.0 Hz, 2H), 7.05 (d, *J* = 10.0 Hz, 2H), 7.02 (d, *J* = 10.0 Hz, 2H), 6.71 (d, *J* = 10.0 Hz, 2H), 4.06 (t, *J* = 7.5 Hz, 2H), 1.72–1.78 (m, 2H), 1.42–1.48 (m, 2H), 1.26–1.38 (m, 12H), 0.87 (t, *J* = 5.0 Hz, 3H), one exchangeable proton was not observed. ^13^C-NMR (125 MHz, DMSO-d_6_) δ, ppm: 166.54, 160.69, 152.14, 147.54, 144.31, 137.50, 131.57, 129.06, 124.77, 123.77, 120.79, 115.65, 114.37, 68.84, 31.61, 29.28, 29.24, 29.18, 29.07, 28.94, 25.88, 22.31, 14.02. ^31^P-NMR (500 MHz, DMSO-d_6_) δ, ppm: 7.77 (s, 1P). CHN elemental analysis: Calculated for C_174_H_204_N_21_O_18_P_3_: C: 70.35%, H: 6.92%, N: 9.90%; Found: C: 70.09%, H: 6.88%, N: 9.84%.

Hexakis{4-((*E*)-(4-((*E*)-4-dodecyloxy-phenyl)diazenyl)phenyl)benzamide}triazaphosphazene, **6d**: Yield: 2.25 g (68.68%), mp: 140.1–142.8 °C, red powder. FTIR (cm^−1^): 3344 (N-H stretching), 2921 and 2849 (C*sp^3^*-H stretching), 1698 (C=O stretching), 1591 (aromatic C=C stretching), 1485 (N=N stretching), 1211 (C-O stretching), 1186 (P=N stretching), 1154 (C-N stretching), 1017 (P-O-C stretching). ^1^H-NMR (500 MHz, DMSO-d_6_) δ, ppm: 7.84 (d, *J* = 10.0 Hz, 2H), 7.70 (d, *J* = 10.0 Hz, 2H), 7.60 (d, *J* = 5.0 Hz, 2H), 7.05 (d, *J* = 10.0 Hz, 2H), 7.02 (d, *J* = 10.0 Hz, 2H), 6.71 (d, *J* = 10.0 Hz, 2H), 4.06 (t, *J* = 7.5 Hz, 2H), 1.72–1.77 (m, 2H), 1.42–1.47 (m, 2H), 1.23–1.38 (m, 16H), 0.86 (t, *J* = 7.5 Hz, 3H), one exchangeable proton was not observed. ^13^C-NMR (125 MHz, DMSO-d_6_) δ, ppm: 166.58, 160.69, 152.12, 147.52, 145.59, 137.13, 131.57, 129.71, 124.77, 123.77, 120.79, 115.65, 114.37, 68.83, 31.60, 29.31, 29.28, 29.24, 29.20, 29.14, 29.03, 28.94, 25.85, 22.30, 14.00. ^31^P-NMR (500 MHz, DMSO-d_6_) δ, ppm: 7.79 (s, 1P). CHN elemental analysis: Calculated for C_186_H_228_N_21_O_18_P_3_: C: 71.17%, H: 7.32%, N: 9.37%; Found: C: 70.88%, H: 7.30%, N: 9.33%.

Hexakis{4-((*E*)-(4-((*E*)-4-tetradecyloxy-phenyl)diazenyl)phenyl)benzamide}triazaphosphazene, **6e**: Yield: 2.55 g (73.88%), mp: 137.9–140.3 °C, red powder. FTIR (cm^−1^): 3342 (N-H stretching), 2916 and 2850 (C*sp^3^*-H stretching), 1703 (C=O stretching), 1593 (aromatic C=C stretching), 1483 (N=N stretching), 1219 (C-O stretching), 1186 (P=N stretching), 1149 (C-N stretching), 1020 (P-O-C stretching). ^1^H-NMR (500 MHz, DMSO-d_6_) δ, ppm: 7.84 (d, *J* = 10.0 Hz, 2H), 7.70 (d, *J* = 10.0 Hz, 2H), 7.60 (d, *J* = 10.0 Hz, 2H), 7.05 (d, *J* = 10.0 Hz, 2H), 7.02 (d, *J* = 10.0 Hz, 2H), 6.71 (d, *J* = 10.0 Hz, 2H), 4.06 (t, *J* = 5.0 Hz, 2H), 1.72–1.77 (m, 2H), 1.42–1.47 (m, 2H), 1.25–1.38 (m, 20H), 0.86 (t, *J* = 7.5 Hz, 3H), one exchangeable proton was not observed. ^13^C-NMR (125 MHz, DMSO-d_6_) δ, ppm: 166.57, 160.64, 152.11, 147.46, 144.21, 135.92, 131.58, 129.03, 124.80, 123.77, 120.81, 115.56 114.35, 68.74, 31.64, 29.37, 29.36, 29.35, 29.33, 29.30, 29.28, 29.16, 29.07, 28.99, 25.88, 22.34, 14.06. ^31^P-NMR (500 MHz, DMSO-d_6_) δ, ppm: 7.80 (s, 1P). CHN elemental analysis: Calculated for C_198_H_252_N_21_O_18_P_3_: C: 71.92%, H: 7.68%, N: 8.89%; Found: C: 71.66%, H: 7.63%, N: 8.81%.

Hexakis{4-((*E*)-(4-((*E*)-4-hydroxy-phenyl)diazenyl)phenyl)benzamide}triazaphosphazene, **6f**: Yield: 1.43 g (64.34%), mp: 260.4–263.1 °C, orange powder. FTIR (cm^−1^): 3344 (N-H stretching), 3200 (O-H stretching), 1701 (C=O stretching), 1590 (aromatic C=C stretching), 1470 (N=N stretching), 1245 (C-O stretching), 1186 (P=N stretching), 1149 (C-N stretching), 1013 (P-O-C stretching). ^1^H-NMR (500 MHz, DMSO-d_6_) δ, ppm: 11.13 (s, 1H), 10.45 (s, 1H), 8.09 (d, *J* = 10.0 Hz, 2H), 7.78 (d, *J* = 10.0 Hz, 4H), 7.56 (d, *J* = 5.0 Hz, 2H), 6.94 (d, *J* = 10.0 Hz, 2H), 6.92 (d, *J* = 10.0 Hz, 2H). ^13^C-NMR (125 MHz, DMSO-d_6_) δ, ppm: 164.36, 161.71, 151.13, 145.69, 145.58, 140.07, 135.30, 129.82, 126.59, 125.48, 124.17, 116.46, 116.23. ^31^P-NMR (500 MHz, DMSO-d_6_) δ, ppm: 7.85 (s, 1P). CHN elemental analysis: Calculated for C_114_H_84_N_21_O_18_P_3_: C: 64.32%, H: 3.98%, N: 13.82%; Found: C: 64.29%, H: 4.00%, N: 13.80%.

Hexakis{4-((*E*)-(4-((*E*)-4-carboxy-phenyl)diazenyl)phenyl)benzamide}triazaphosphazene, **6g**: Yield: 1.45 g (60.46%), mp: 242.7–245.1 °C, dark orange powder. FTIR (cm^−1^): 3343 (N-H stretching), 3205 (O-H stretching), 1687 (C=O stretching), 1591 (aromatic C=C stretching), 1475 (N=N stretching), 1213 (C-O stretching), 1181 (P=N stretching), 1157 (C-N stretching), 1012 (P-O-C stretching). ^1^H-NMR (500 MHz, DMSO-d_6_) δ, ppm: 7.81 (d, *J* = 10.0 Hz, 2H), 7.66 (d, *J* = 5.0 Hz, 2H), 7.60 (d, *J* = 10.0 Hz, 2H), 6.89 (d, *J* = 10.0 Hz, 2H), 6.84 (d, *J* = 10.0 Hz, 2H), 6.67 (d, *J* = 10.0 Hz, 2H), two exchangeable protons were not observed. ^13^C-NMR (125 MHz, DMSO-d_6_) δ, ppm: 167.83, 162.02, 159.63, 152.23, 146.02, 143.42, 132.00, 124.86, 124.28, 124.08, 122.07, 116.18, 115.60, 113.95. ^31^P-NMR (500 MHz, DMSO-d_6_) δ, ppm: 7.83 (s, 1P). CHN elemental analysis: Calculated for C_120_H_84_N_21_O_24_P_3_: C: 62.75%, H: 3.69%, N: 12.81%; Found: C: 62.67%, H: 3.70%, N: 12.73%.

Hexakis{4-((*E*)-(4-((*E*)-4-chloro-phenyl)diazenyl)phenyl)benzamide}triazaphosphazene, **6h**: Yield: 1.44 g (61.58%), mp: 233.3–236.1 °C, orange powder. FTIR (cm^−1^): 3340 (N-H stretching), 1702 (C=O stretching), 1596 (aromatic C=C stretching), 1471 (N=N stretching), 1222 (C-O stretching), 1187 (P=N stretching), 1147 (C-N stretching), 1017 (P-O-C stretching), 811 (C-Cl bending). ^1^H-NMR (500 MHz, DMSO-d_6_) δ, ppm: 8.03 (d, *J* = 10.0 Hz, 2H), 7.69 (d, *J* = 10.0 Hz, 2H), 7.68 (d, *J* = 5.0 Hz, 2H), 7.64 (d, *J* = 10.0 Hz, 2H), 6.70 (d, *J* = 10.0 Hz, 2H), 6.56 (d, *J* = 10.0 Hz, 2H), one exchangeable proton was not observed. ^13^C-NMR (125 MHz, DMSO-d_6_) δ, ppm: 170.08, 169.01, 153.67, 153.33, 152.94, 143.49, 139.49, 131.62, 130.53, 125.73, 121.36, 114.01, 113.12. ^31^P-NMR (500 MHz, DMSO-d_6_) δ, ppm: 7.86 (s, 1P). CHN elemental analysis: Calculated for C_114_H_78_Cl_6_N_21_O_12_P_3_: C: 61.14%, H: 3.51%, N: 13.13%; Found: C: 60.73%, H: 3.49%, N: 13.01%. 

Hexakis{4-((*E*)-(4-((*E*)-4-nitro-phenyl)diazenyl)phenyl)benzamide}triazaphosphazene, **6i**: Yield: 2.84 g (78.75%), mp: 268.7–270.1 °C, brown powder. FTIR (cm^−1^): 3343 (N-H stretching), 1700 (C=O stretching), 1590 (aromatic C=C stretching), 1475 (N=N stretching), 1213 (C-O stretching), 1184 (P=N stretching), 1160 (C-N stretching), 1017 (P-O-C stretching). ^1^H-NMR (500 MHz, DMSO-d_6_) δ, ppm: 8.25 (d, *J* = 5.0 Hz, 2H), 7.88 (d, *J* = 10.0 Hz, 2H), 7.85 (d, *J* = 10.0 Hz, 2H), 7.78 (d, *J* = 10.0 Hz, 2H), 6.95 (d, *J* = 5.0 Hz, 2H), 6.63 (d, *J* = 5.0 Hz, 2H), one exchangeable proton was not observed. ^13^C-NMR (125 MHz, DMSO-d_6_) δ, ppm: 162.49, 156.35, 155.70, 148.40, 146.32, 140.33, 137.15, 126.43, 125.92, 125.01, 123.16, 116.68, 113.21. ^31^P-NMR (500 MHz, DMSO-d_6_) δ, ppm: 7.81 (s, 1P). CHN elemental analysis: Calculated for C_114_H_78_N_27_O_24_P_3_: C: 59.46%, H: 3.41%, N: 16.42%; Found: C: 58.98%, H: 3.38%, N: 16.27%.

#### 3.3.9. Synthesis of Hexakis{4-((E)-(4-((E)-4-amino-phenyl)diazenyl)phenyl)benzamide} triazaphosphazene, **6j**

A solution of compound **6i** (1.60 g, 0.70 mmol) in 20 mL of hot ethanol and a solution of sodium sulphide hydrate, Na_2_S.9H_2_O (0.54 g, 6.95 mmol) in 20 mL of ethanol and 20 mL of distilled water were mixed in a 100 mL round bottom flask. The reaction mixture was refluxed for 24 h. The reaction progress was monitored by TLC. Upon completion, the mixture was cooled in ice water and the precipitate formed was filtered, washed with cold ethanol and dried overnight in open air. Yield: 1.11 g (75.26%), mp: 285.5–287.4 °C, brown powder. FTIR (cm^−1^): 3340 (N-H amide stretching), 3381, and 3218 (N-H amine stretching), 1687 (C=O stretching), 1601 (aromatic C=C stretching), 1490 (N=N stretching), 1211 (C-O stretching), 1186 (P=N stretching), 1158 (C-N stretching), 1013 (P-O-C stretching). ^1^H-NMR (500 MHz, DMSO-d_6_) δ, ppm: 10.46 (s, 1H), 7.80 (d, *J* = 10.0 Hz, 4H), 7.57 (d, *J* = 10.0 Hz, 2H), 7.00 (d, *J* = 10.0 Hz, 2H), 6.96 (d, *J* = 10.0 Hz, 2H), 6.57 (d, *J* = 5.0 Hz, 2H), 5.20 (s, 2H). ^13^C-NMR (125 MHz, DMSO-d_6_) δ, ppm: 161.75, 151.21, 151.15, 148.05, 145.59, 135.31, 129.85, 128.95, 125.50, 124.20, 119.33, 116.49, 115.73. ^31^P-NMR (500 MHz, DMSO-d_6_) δ, ppm: 7.80 (s, 1P). CHN elemental analysis: Calculated for C_114_H_90_N_27_O_12_P_3_: C: 64.49%, H: 4.27%, N: 17.81%; Found: C: 64.11%, H: 4.23%, N: 17.77%.

#### 3.3.10. Synthesis of Hexakis(4-nitro-phenoxy)cyclotriphosphazene, **7**

4-Nitrophenol (14.61 g, 0.11 mol), phosphonitrilic chloride trimer (5.22 g, 0.015 mol), and potassium carbonate, K_2_CO_3_ (20.73 g, 0.15 mol) were mixed in 150 mL of acetone in a 250 mL round bottom flask. The mixture was refluxed for 4 days. The reaction progress was monitored using TLC. Upon completion, the mixture was poured into 250 mL of cool water. The precipitate formed was filtered and dried overnight in open air. Yield: 12.15 g (84.11%), mp: 182.5–185.1 °C, yellow powder. FTIR (cm^−1^): 1603 (aromatic C=C stretching), 1256 (C-O stretching), 1182 (P=N stretching), 1160 (C-N stretching), 977 (P-O-C stretching). ^1^H-NMR (500 MHz, DMSO-d_6_) δ, ppm: 8.13 (d, *J* = 10.0 Hz, 2H), 7.30 (d, *J* = 10.0 Hz, 2H). ^13^C-NMR (125 MHz, DMSO-d_6_) δ, ppm: 154.34, 145.68, 126.05, 121.89. ^31^P-NMR (500 MHz, DMSO-d_6_) δ, ppm: 7.21 (s, 1P). CHN elemental analysis: Calculated for C_36_H_24_N_9_O_18_P_3_: C: 44.88%, H: 13.08%, N: 13.08%; Found: C: 44.67%, H: 12.99%, N: 13.03%.

#### 3.3.11. Synthesis of Hexakis{4-((E)-((4-((E)-4-substituted-phenyl)diazenyl)phenyl)diazenyl)phenoxy} triazaphosphazene, **8a–i**

Hexakis{4-((*E*)-((4-((*E*)-4-heptyloxy-phenyl)diazenyl)phenyl)diazenyl)phenoxy} triazaphosphazene, **8a**: Intermediate **7** (1.00 g, 1.04 mmol) and intermediate **3a** (2.26 g, 7.27 mmol) were mixed in 50 mL ethanol and the solution was poured into a 100 mL round bottom flask. 10 mL of 3% aqueous NaOH was added and the mixture was refluxed for 24 h. The reaction progress was monitored by TLC. Upon completion, the mixture was cooled into an ice bath. The precipitate formed was filtered, washed with cold ethanol, and dried overnight in open air to give a red precipitate. The same method was used to synthesize **8b****–i**. Yield: 1.85 g (67.77%), mp: 116.9–118.5 °C, red powder. FTIR (cm^−1^): 2921 and 2860 (C*sp^3^*-H stretching), 1603 (aromatic C=C stretching), 1496 (N=N stretching), 1248 (C-O stretching), 1189 (P=N stretching), 1156 (C-N stretching), 997 (P-O-C stretching). ^1^H-NMR (500 MHz, DMSO-d_6_) δ, ppm: 8.23 (d, *J* = 10.0 Hz, 2H), 8.15 (d, *J* = 10.0 Hz, 2H), 7.76 (d, *J* = 10.0 Hz, 2H), 7.72 (d, *J* = 5.0 Hz, 2H), 7.02 (d, *J* = 10.0 Hz, 2H), 6.94 (d, *J* = 10.0 Hz, 2H), 4.03 (t, *J* = 7.5 Hz, 2H), 1.70–1.75 (m, 2H), 1.39–1.44 (m, 2H), 1.23–1.33 (m, 6H), 0.85 (t, *J* = 7.5 Hz, 3H). ^13^C-NMR (125 MHz, DMSO-d_6_) δ, ppm: 166.40, 161.26, 160.72, 150.47, 147.16, 146.26, 130.90, 124.55, 124.19, 123.66, 116.35, 115.50, 68.75, 31.52, 29.13, 28.71, 25.82, 22.26, 13.96. ^31^P-NMR (500 MHz, DMSO-d_6_) δ, ppm: 11.16 (s, 1P). CHN elemental analysis: Calculated for C_150_H_162_N_27_O_12_P_3_: C: 68.55%, H: 6.21%, N: 14.39%; Found: C: 68.18%, H: 6.17%, N: 14.23%.

Hexakis{4-((*E*)-((4-((*E*)-4-nonyloxy-phenyl)diazenyl)phenyl)diazenyl)phenoxy} triazaphosphazene, **8b**: Yield: 2.10 g (72.41%), mp: 116.1–118.3 °C, dark red powder. FTIR (cm^−1^): 2921 and 2851 (C*sp^3^*-H stretching), 1601 (aromatic C=C stretching), 1496 (N=N stretching), 1248 (C-O stretching), 1181 (P=N stretching), 1151 (C-N stretching), 998 (P-O-C stretching). ^1^H-NMR (500 MHz, DMSO-d_6_) δ, ppm: 8.23 (d, *J* = 10.0 Hz, 2H), 8.15 (d, *J* = 10.0 Hz, 2H), 7.76 (d, *J* = 10.0 Hz, 2H), 7.72 (d, *J* = 5.0 Hz, 2H), 7.02 (d, *J* = 10.0 Hz, 2H), 6.94 (d, *J* = 10.0 Hz, 2H), 4.03 (t, *J* = 7.5 Hz, 2H), 1.70–1.75 (m, 2H), 1.39–1.44 (m, 2H), 1.22–1.35 (m, 10H), 0.85 (t, *J* = 7.5 Hz, 3H). ^13^C-NMR (125 MHz, DMSO-d_6_) δ, ppm: 166.52, 161.27, 160.78, 150.32, 147.18, 146.25, 130.87, 124.56, 124.20, 123.61, 116.36, 115.54, 68.76, 31.61, 29.25, 29.14, 29.10, 28.91, 25.87, 22.33, 14.01. ^31^P-NMR (500 MHz, DMSO-d_6_) δ, ppm: 11.19 (s, 1P). CHN elemental analysis: Calculated for C_162_H_186_N_27_O_12_P_3_: C: 69.58%, H: 6.70%, N: 13.52%; Found: C: 69.44%, H: 6.68%, N: 13.44%.

Hexakis{4-((*E*)-((4-((*E*)-4-decyloxy-phenyl)diazenyl)phenyl)diazenyl)phenoxy} triazaphosphazene, **8c**: Yield: 2.18 g (72.91%), mp: 113.3–115.1 °C, orange powder. FTIR (cm^−1^): 2919 and 2851 (C*sp^3^*-H stretching), 1603 (aromatic C=C stretching), 1493 (N=N stretching), 1246 (C-O stretching), 1178 (P=N stretching), 1146 (C-N stretching), 997 (P-O-C stretching). ^1^H-NMR (500 MHz, DMSO-d_6_) δ, ppm: 8.22 (d, *J* = 10.0 Hz, 2H), 8.14 (d, *J* = 10.0 Hz, 2H), 7.74 (d, *J* = 10.0 Hz, 2H), 7.70 (d, *J* = 10.0 Hz, 2H), 6.99 (d, *J* = 10.0 Hz, 2H), 6.93 (d, *J* = 10.0 Hz, 2H), 4.01 (t, *J* = 5.0 Hz, 2H), 1.68–1.73 (m, 2H), 1.37–1.42 (m, 2H), 1.19–1.32 (m, 12H), 0.82 (t, *J* = 5.0 Hz, 3H). ^13^C-NMR (125 MHz, DMSO-d_6_) δ, ppm: 166.47, 161.25, 160.73, 150.41, 147.17, 146.26, 130.89, 124.53, 124.18, 123.62, 116.33, 115.48, 68.74, 31.62, 29.28, 29.25, 29.12, 29.08, 28.96, 25.85, 22.32, 13.96. ^31^P-NMR (500 MHz, DMSO-d_6_) δ, ppm: 11.18 (s, 1P). CHN elemental analysis: Calculated for C_168_H_198_N_27_O_12_P_3_: C: 70.05%, H: 6.93%, N: 13.13%; Found: C: 69.89%, H: 6.90%, N: 13.02%.

Hexakis{4-((*E*)-((4-((*E*)-4-dodecyloxy-phenyl)diazenyl)phenyl)diazenyl)phenoxy} triazaphosphazene, **8d**: Yield: 2.20 g (69.62%), mp: 109.7–111.4 °C, dark orange powder. FTIR (cm^−1^): 2916 and 2851 (C*sp^3^*-H stretching), 1601 (aromatic C=C stretching), 1493 (N=N stretching), 1248 (C-O stretching), 1192 (P=N stretching), 1151 (C-N stretching), 995 (P-O-C stretching). ^1^H-NMR (500 MHz, DMSO-d_6_) δ, ppm: 8.21 (d, *J* = 5.0 Hz, 2H), 8.14 (d, *J* = 10.0 Hz, 2H), 7.74 (d, *J* = 10.0 Hz, 2H), 7.70 (d, *J* = 5.0 Hz, 2H), 7.01 (d, *J* = 10.0 Hz, 2H), 6.94 (d, *J* = 5.0 Hz, 2H), 4.05 (t, *J* = 5.0 Hz, 2H), 1.71–1.76 (m, 2H), 1.41–1.46 (m, 2H), 1.23–1.36 (m, 16H), 0.85 (t, *J* = 7.5 Hz, 3H). ^13^C-NMR (125 MHz, DMSO-d_6_) δ, ppm: 166.57, 161.33, 160.76, 150.42, 147.40, 146.45, 130.81, 124.46, 124.14, 123.49, 116.40, 115.65, 68.94, 31.60, 29.38, 29.30, 29.29, 29.24, 29.16, 29.05, 28.94, 25.85, 22.26, 13.85. ^31^P-NMR (500 MHz, DMSO-d_6_) δ, ppm: 11.15 (s, 1P). CHN elemental analysis: Calculated for C_180_H_222_N_27_O_12_P_3_: C: 70.91%, H: 7.34%, N: 12.40%; Found: C: 70.66%, H: 7.33%, N: 12.33%.

Hexakis{4-((*E*)-((4-((*E*)-4-tetradecyloxy-phenyl)diazenyl)phenyl)diazenyl)phenoxy} triazaphosphazene, **8e**: Yield: 2.55 g (76.35%), mp: 115.9–117.5 °C, orange powder. FTIR (cm^−1^): 2916 and 2851 (C*sp^3^*-H stretching), 1603 (aromatic C=C stretching), 1493 (N=N stretching), 1251 (C-O stretching), 1176 (P=N stretching), 1149 (C-N stretching), 995 (P-O-C stretching). ^1^H-NMR (500 MHz, DMSO-d_6_) δ, ppm: 8.22 (d, *J* = 10.0 Hz, 2H), 8.15 (d, *J* = 10.0 Hz, 2H), 7.75 (d, *J* = 10.0 Hz, 2H), 7.70 (d, *J* = 10.0 Hz, 2H), 7.01 (d, *J* = 5.0 Hz, 2H), 6.93 (d, *J* = 5.0 Hz, 2H), 4.04 (t, *J* = 5.0 Hz, 2H), 1.71–1.76 (m, 2H), 1.40–1.46 (m, 2H), 1.22–1.37 (m, 20H), 0.85 (t, *J* = 5.0 Hz, 3H). ^13^C-NMR (125 MHz, DMSO-d_6_) δ, ppm: 166.47, 161.30, 160.72, 150.52, 147.33, 146.40, 130.90, 124.48, 124.14, 123.57, 116.34, 115.55, 68.85, 31.62, 29.41, 29.37, 29.36, 29.35, 29.33, 29.27, 29.17, 29.08, 28.97, 25.87, 22.30, 13.87. ^31^P-NMR (500 MHz, DMSO-d_6_) δ, ppm: 11.17 (s, 1P). CHN elemental analysis: Calculated for C_192_H_246_N_27_O_12_P_3_: C: 71.68%, H: 7.71%, N: 11.76%; Found: C: 71.22%, H: 7.67%, N: 11.68%.

Hexakis{4-((*E*)-((4-((*E*)-4-hydroxy-phenyl)diazenyl)phenyl)diazenyl)phenoxy} triazaphosphazene, **8f**: Yield: 1.28 g (60.38%), mp: 180.7–182.9 °C, dark orange powder. FTIR (cm^−1^): 3255 (O-H stretching), 1601 (aromatic C=C stretching), 1496 (N=N stretching), 1252 (C-O stretching), 1178 (P=N stretching), 1150 (C-N stretching), 1017 (P-O-C stretching). ^1^H-NMR (500 MHz, DMSO-d_6_) δ, ppm: 8.10 (d, *J* = 5.0 Hz, 2H), 7.94 (d, *J* = 5.0 Hz, 2H), 7.84 (d, *J* = 10.0 Hz, 2H), 7.82 (d, *J* = 10.0 Hz, 2H), 6.96 (d, *J* = 10.0 Hz, 2H), 6.61 (d, *J* = 10.0 Hz, 2H), one exchangeable proton was not observed. ^13^C-NMR (125 MHz, DMSO-d_6_) δ, ppm: 167.37, 162.03, 156.05, 155.04, 145.82, 136.22, 130.98, 126.81, 125.76, 122.50, 116.51, 112.87. ^31^P-NMR (500 MHz, DMSO-d_6_) δ, ppm: 11.19 (s, 1P). CHN elemental analysis: Calculated for C_108_H_78_N_27_O_12_P_3_: C: 63.62%, H: 3.86%, N: 18.55%; Found: C: 63.50%, H: 3.90%, N: 18.39%.

Hexakis{4-((*E*)-((4-((*E*)-4-carboxy-phenyl)diazenyl)phenyl)diazenyl)phenoxy} triazaphosphazene, **8g**: Yield: 1.36 g (59.39%), mp: 190.1–192.9 °C, orange powder. FTIR (cm^−1^): 3251 (O-H stretching), 1684 (C=O stretching), 1603 (aromatic C=C stretching), 1493 (N=N stretching), 1251 (C-O stretching), 1176 (P=N stretching), 1148 (C-N stretching), 1007 (P-O-C stretching). ^1^H-NMR (500 MHz, DMSO-d_6_) δ, ppm: 8.10 (d, *J* = 10.0 Hz, 2H), 7.85 (d, *J* = 10.0 Hz, 2H), 7.83 (d, *J* = 10.0 Hz, 2H), 6.97 (d, *J* = 10.0 Hz, 2H), 6.54 (d, *J* = 10.0 Hz, 2H), 6.51 (d, *J* = 5.0 Hz, 2H), one exchangeable proton was not observed. ^13^C-NMR (125 MHz, DMSO-d_6_) δ, ppm: 167.51, 162.08, 154.97, 149.41, 145.80, 132.72, 131.57, 130.99, 125.77, 122.51, 116.55, 116.10, one carbon environments not observed due to coincident resonances. ^31^P-NMR (500 MHz, DMSO-d_6_) δ, ppm: 11.23 (s, 1P). CHN elemental analysis: Calculated for C_114_H_78_N_27_O_18_P_3_: C: 62.04%, H: 3.56%, N: 17.14%; Found: C: 61.79%, H: 3.55%, N: 17.02%.

Hexakis{4-((*E*)-((4-((*E*)-4-chloro-phenyl)diazenyl)phenyl)diazenyl)phenoxy}triazaphosphazene, **8h**: Yield: 1.57 g (70.40%), mp: 177.5–178.1 °C, yellow powder. FTIR (cm^−1^): 1606 (aromatic C=C stretching), 1491 (N=N stretching), 1255 (C-O stretching), 1180 (P=N stretching), 1149 (C-N stretching), 996 (P-O-C stretching), 790 (C-Cl bending). ^1^H-NMR (500 MHz, DMSO-d_6_) δ, ppm: 8.09 (d, *J* = 10.0 Hz, 2H), 7.91 (d, *J* = 5.0 Hz, 2H), 7.83 (d, *J* = 5.0 Hz, 2H), 7.81 (d, *J* = 10.0 Hz, 2H), 7.46 (d, *J* = 5.0 Hz, 2H), 6.95 (d, *J* = 10.0 Hz, 2H). ^13^C-NMR (125 MHz, DMSO-d_6_) δ, ppm: 167.33, 166.97, 162.04, 155.04, 145.82, 138.29, 131.53, 130.96, 129.05, 125.75, 122.49, 116.49. ^31^P-NMR (500 MHz, DMSO-d_6_) δ, ppm: 11.20 (s, 1P). CHN elemental analysis: Calculated for C_108_H_72_Cl_6_N_27_O_6_P_3_: C: 60.35%, H: 3.38%, N: 17.59%; Found: C: 60.11%, H: 3.35%, N: 17.40%.

Hexakis{4-((*E*)-((4-((*E*)-4-nitro-phenyl)diazenyl)phenyl)diazenyl)phenoxy}triazaphosphazene, **8i**: Yield: 1.68 g (73.04%), mp: 193.5–195.3 °C, red powder. FTIR (cm^−1^): 1603 (aromatic C=C stretching), 1496 (N=N stretching), 1253 (C-O stretching), 1179 (P=N stretching), 1148 (C-N stretching), 1006 (P-O-C stretching). ^1^H-NMR (500 MHz, DMSO-d_6_) δ, ppm: 7.94 (d, *J* = 5.0 Hz, 2H), 7.65 (d, *J* = 10.0 Hz, 2H), 7.61 (d, *J* = 5.0 Hz, 2H), 7.48 (d, *J* = 10.0 Hz, 2H), 6.92 (d, *J* = 5.0 Hz, 2H), 6.72 (d, *J* = 5.0 Hz, 2H). ^13^C-NMR (125 MHz, DMSO-d_6_) δ, ppm: 166.94, 159.70, 151.84, 146.58, 144.31, 138.20, 131.44, 128.94, 124.63, 123.97, 116.29, 114.34. ^31^P-NMR (500 MHz, DMSO-d_6_) δ, ppm: 11.18 (s, 1P). CHN elemental analysis: Calculated for C_108_H_72_N_33_O_18_P_3_: C: 58.62%, H: 3.28%, N: 20.89%; Found: C: 58.66%, H: 3.25%, N: 20.65%.

#### 3.3.12. Synthesis of Hexakis(4-acetamido-phenoxy)cyclotriphosphazene, **9**

4-Acetamidophenol (10.57 g, 0.07 mol), phosphonitrilic chloride trimer (3.48 g, 0.01 mol), and potassium carbonate, K_2_CO_3_ (13.82 g, 0.1 mol) were mixed in 150 mL of acetone and the solution was transferred into a 250 mL round bottom flask. The reaction mixture was refluxed for 4 days. The reaction progress was monitored using TLC. Upon completion, the mixture was poured into a 250 mL of cooled water. The precipitate formed was filtered using Buchner funnel, washed with cold water, and then dried overnight in open air to give a white solid powder. Yield: 9.15 g (88.41%), mp: 178.8–180.7 °C, white powder. FTIR (cm^−1^): 3343 (N-H stretching), 2890 and 2821 (C*sp^3^*-H stretching), 1703 (C=O stretching), 1602 (aromatic C=C stretching), 1253 (C-O stretching), 1179 (P=N stretching), 1155 (C-N stretching), 971 (P-O-C stretching). ^1^H-NMR (500 MHz, DMSO-d_6_) δ, ppm: 9.92 (s, 1H), 7.44 (d, *J* = 10.0 Hz, 2H), 6.80 (d, *J* = 10.0 Hz, 2H), 2.04 (s, 3H). ^13^C-NMR (125 MHz, DMSO-d_6_) δ, ppm: 168.35, 145.09, 136.32, 120.52, 120.19, 23.80. ^31^P-NMR (500 MHz, DMSO-d_6_) δ, ppm: 9.18 (s, 1P). CHN elemental analysis: Calculated for C_48_H_48_N_9_O_12_P_3_: C: 55.66%, H: 4.67%, N: 12.17%; Found: C: 55.37%, H: 4.63%, N: 12.10%.

#### 3.3.13. Synthesis of Hexakis(4-amino-phenoxy)cyclotriphosphazene, **10**

A solution of intermediate **9** (8.00 g, 7.73 mmol) in 100 mL methanol and a solution of sodium hydroxide, NaOH (4.64 g, 0.12 mol) in 20 mL water were mixed in a 250 mL round bottom flask. The mixture was refluxed for 24 h. The reaction progress was monitored using TLC. Upon completion, the mixture was poured into a 250 mL of cooled water. The precipitate formed was filtered, washed with cold water, and dried overnight in open air to give a white precipitate. Yield: 4.80 g (79.34%), mp: 190.1–192.9 °C, white powder. FTIR (cm^−1^): 3451 and 3217 (N-H stretching), 1606 (aromatic C=C stretching), 1258 (C-O stretching), 1181 (P=N stretching), 1162 (C-N stretching), 970 (P-O-C stretching). ^1^H-NMR (500 MHz, DMSO-d_6_) δ, ppm: 6.52 (d, *J* = 10.0 Hz, 2H), 6.44 (d, *J* = 5.0 Hz, 2H), 4.89 (s, 2H). ^13^C-NMR (125 MHz, DMSO-d_6_) δ, ppm: 145.56, 140.83, 120.91, 114.26. ^31^P-NMR (500 MHz, DMSO-d_6_) δ, ppm: 14.73 (s, 1P). CHN elemental analysis: Calculated for C_36_H_36_N_9_O_6_P_3_: C: 55.18%, H: 4.63%, N: 16.09%; Found: C: 54.78%, H: 4.60%, N: 16.10%.

#### 3.3.14. Synthesis of Hexakis{4-((E)-((4-((E)-4-amino-phenyl)diazenyl)phenyl)diazenyl)phenoxy} triazaphosphazene, **8j**


Intermediate **10** (1.00 g, 1.28 mmol) and intermediate **3i** (2.16 g, 8.94 mmol) were used in the reaction. Synthetic protocol for **8a** was used in this reaction. Yield: 1.31 g (51.98%), mp: 197.7–199.6 °C, orange powder. FTIR (cm^−1^): 3385 and 3272 (N-H stretching), 1601 (aromatic C=C stretching), 1490 (N=N stretching), 1251 (C-O stretching), 1176 (P=N stretching), 1152 (C-N stretching), 1013 (P-O-C stretching). ^1^H-NMR (500 MHz, DMSO-d_6_) δ, ppm: 8.03 (d, *J* = 5.0 Hz, 2H), 8.02 (d, *J* = 10.0 Hz, 2H), 7.70 (d, *J* = 10.0 Hz, 2H), 7.68 (d, *J* = 5.0 Hz, 2H), 6.79 (d, *J* = 5.0 Hz, 2H), 6.69 (d, *J* = 10.0 Hz, 2H), 6.13 (s, 2H). ^13^C-NMR (125 MHz, DMSO-d_6_) δ, ppm: 169.83, 169.05, 153.75, 153.40, 143.46, 137.28, 130.51, 126.85, 125.74, 121.37, 117.12, 113.95. ^31^P-NMR (500 MHz, DMSO-d_6_) δ, ppm: 11.22 (s, 1P). CHN elemental analysis: Calculated for C_108_H_84_N_33_O_6_P_3_: C: 63.81%, H: 4.16%, N: 22.74%; Found: C: 63.44%, H: 4.18%, N: 22.67%.

## 4. Conclusions

All the intermediates and hexasubstituted cyclotriphosphazene compounds with amide-azo (**6a****–j**) and azo-azo (**8a****–j**) linking units were successfully synthesized and characterized. The POM observation showed only intermediates **2a****–e** with nitro terminal group exhibited smectic A. Meanwhile, only compounds **6a****–e** (Series 1) and **8a****–e** (Series 2) with the heptyl, nonyl, decyl, dodecyl, and tertradecyl terminal chains, respectively, displayed the properties of liquid crystals of smectic phases. Compounds **6a**,**b** showed the smectic A phase, while compounds **6c**,**d** and **8a****–e** exhibited smectic C phase. However, all the compounds with a small substituent, such as hydroxyl, carboxyl, chloro, nitro, and amino terminal substituents, were found to be non-mesogenic without any liquid crystal behaviour. In addition, the fire-retardant properties of final compounds in Series 1 and 2 showed positive results in terms of the LOI values. Compound **6i** has the highest LOI value compared to other homologue series. The effect of the electron withdrawing nitro group induced the compounds to have high fire retardancy.

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
