# Peer review of "Synthesis of New Star-Shaped Liquid Crystalline Cyclotriphosphazene Derivatives with Fire Retardancy Bearing Amide-Azo and Azo-Azo Linking Units"

_ijms, 2020, doi:10.3390/ijms21124267_

Round 1

Reviewer 1 Report

The manuscript “Synthesis of new star-shaped liquid crystalline cyclotriphosphazene derivatives with fire retardancy bearing amide-azo and azo-azo linking units” by Zuhair Jamain, Melati Khairuddean and Tay Guan-Seng (ijms-824539) represents very large research work. This is especially true for the synthetic part of the work including multistep organic synthesis.

All substances were fully characterized by IR, NMR, CHN, POM, and DSC methods.  It is very sad that manuscript does not contain any structural information from single crystal investigation.

The manuscript left a good impression and deserves to be published after some a moderate correction.

There are several comments on the experimental part.

1)  In some case, the synthetic details are not complete (see the comments in the pdf file of the manuscript).

The weakest point of the article is the low quality of text editing.

First of all, it is a poor and careless writing style. However, there are also methodological problems. Therefore, for example, when describing the properties of the obtained compounds, the authors use the past tense, and exclusively past simple. Nevertheless, if the study was completed, the materials did not cease to possess their chemical and physical properties.

I hope that my comments (see the comments in the pdf file of the manuscript) will help the authors not only significantly improve the text of the manuscript, but also write many good scientific texts in the future.

Author Response

Response to Reviewer Comments

Point 1: In some case, the synthetic details are not complete (see the comments in the pdf file of the manuscript).

Response 1: All the correction have been made according to the comments on the pdf file of the manuscript.

Point 2: The weakest point of the article is the low quality of text editing.

First of all, it is a poor and careless writing style. However, there are also methodological problems. Therefore, for example, when describing the properties of the obtained compounds, the authors use the past tense, and exclusively past simple. Nevertheless, if the study was completed, the materials did not cease to possess their chemical and physical properties.

Response 2: All the correction have been made according to the suggestion in the review report (pdf file of the manuscript).

Reviewer 2 Report

The reviewed article: Synthesis of new star-shaped liquid crystalline cyclotriphosphazene derivatives with fire retardancy bearing amide-azo and azo-azo linking units by Zuhair Jamain, Melati Khairuddean and Tay Guan-Seng presents the synthesis and characteristics of new hexasubstituted cyclotriphosphazene derivatives, which was differentiated by two types of linking units in molecules an amide-azo and an azo-azo.These compounds differed in other terminal substituents such as heptyl, nonyl, decyl, dodecyl, tetradecyl, hydroxycarboxyl, chloro-, nitro- and amino- groups.

 The authors presented the use of synthesized compounds as flame retardants in the case of modified polyester resin.

 In my opinion, the reviewer manuscript introduces a thorough research workshop of the authors and the achieved results exhibit considerable application potential, nevertheless, I have some comments and suggestions:

 • Please specify the conditions for DSC measurements

 • Please specify the FTIR methodology

 • The authors used polyester resin for which basic characteristics are missing

 • Please explain/clarify the statement (page 23 line 475-477): The existence of the hydrogen bonding was reflected in increased thermal stability of the compounds and thus they demonstrated the flexibility and adhesion resistance towards combustion

 • Thermal stability is not the same as fire-retardant properties

 • The mechanical properties of modified materials have also not been tested by the authors

Author Response

Response to Reviewer 1 Comments

Point 1: Please specify the conditions for DSC measurements.

Response 1: Line 485: The conditions for DSC measurements was added in Section 3.2.

Point 2: Please specify the FTIR methodology

Response 2: Line 485: the FTIR methodology was added in Section 3.2.

Point 3: The authors used polyester resin for which basic characteristics are missing

Response 3: In this study, the polyester resin only characterized using LOI testing as this resin only used as a medium for moulding. The LOI data obtained were compared with the synthesized final compounds.

Point 4:

(i) Please explain/clarify the statement (page 23 line 475-477): The existence of the hydrogen bonding was reflected in increased thermal stability of the compounds and thus they demonstrated the flexibility and adhesion resistance towards combustion

(i) Thermal stability is not the same as fire-retardant properties

Response 4: Line 456-461: The existence of the hydrogen bonding was reflected in increased thermal stability and fire retardancy of the compounds. This molecule was transformed into a cross-linked structure, which caused an increase in the heat resistance in the laminate structure. Thus, these compounds able to demonstrate the flexibility and adhesion resistance towards combustion. The higher the thermal stability of compounds, the less combustible the target materials.

Point 5: The mechanical properties of modified materials have also not been tested by the authors

Response 5: This research only focused on both the liquid crystal and fire retardant properties. The interest of this research is to gain a better insight of the structure-properties relationship of these types of compounds on liquid crystal and fire retardant. Thus, the mechanical properties is not included in this work.

Reviewer 3 Report

The authors present an interesting study of hexaaryloxy-substituted cyclotriphosphazene derivatives, which they have been tested for liquid crystal poroperties and fire retardency properties. The structures of final compounds and intermediates have been well characterised via a combination of 1H NMR, 13C NMR, 31P NMR (where appropriate), IR and elemental analysis. The final compounds showed liquid crystal properties, displaying either smectic A or smectic B phases. The compounds showed promising fire retardency properties, with compound 6i showing highest LOI value.

Some language corrections are required in places.

It would have been nice to see mass spectral data, but this is compensated by the elemental analysis data.

I would recommend this manuscript for publication in IJMS, after the following queries have been addressed:

1) Line 21. '...mesogenic with smectic A and C phases' should become 'mesogeneic with either smectic A or C phases'

2) Line 60. 'polymers liquid crystals' should become 'polymer liquid crystals'

3) Line 97.  'which was then reduced to give' should be changed to 'which was then hydrolysed to give', as the conversion of 4 to 5 is the hydrolysis of an ester to an acid, which are at the same oxidation state. 

4) Section 2.2 and 2.3. Here the authors provide a very detailed and solid rationale for their deduction of the chemical structures from the spectral information. Although the information is correct, I feel that this is perhaps 'over-kill', and that a lot of the information here could be moved to the SI and replaced with a sentence like 'the chemical structures of all compounds were consistent with the spectral data'.

5) Line 150. surrounding should be replaced by surroundings.

6) Line 156. 'Inserted into' should be replaced by 'attached to'

7) Line 160. 'with a different synthesis route' could be changed to 'by a different synthetic route'

8) Line 168. 'which confirming' should be just 'confirming'

9) Line 396. 'the SmA phase was disappeared' should become 'the SmA phase disappeared.

10) Line 427. 'Chlorine is polar substituents possessing...' should become 'Chloride is a polar substituent possessing...'

11) Line 562. 2e 13C NMR data, two carbon environments are missing (20 instead of 22). If this is due to resonance overlap, then should write '2 carbon environments not observed due to coincident resonances', or similar. 

12) Line 576. 3a 13C NMR data, one carbon environment is missing (14 instead of 15). If this is due to resonance overlap, then should write 'one carbon environments is not observed due to coincident resonances', or similar.

13) Line 584. 3b 13C NMR data, one carbon environment is missing (16 instead of 17). If this is due to resonance overlap, then should write 'one carbon environments not observed due to coincident resonances', or similar.

14) Line 871. 8e 13C NMR data, one carbon environment too many (27 instead of 26).

15) Line 881. 8g. 13C NMR data, one carbon environment is missing (12 instead of 13. If this is due to resonance overlap, then should write 'one carbon environments not observed due to coincident resonances', or similar.

16) After 1H NMR data for compounds 3f, 5, 6a, 6b, 3g, 6h, 6i, 8f, 8g should include the sentence 'one exchangeable proton was not observed', or similar.

17. After the 1H NMR data for compound 6g, should include the sentence 'two exchangeable protons were not observed'.

18) Line 624. In section 3.2.4, the reaction consists of 200 mL solvent + >35 g reagents in a 250 mL flask. Safety-wise, this seems rather full to allow for bubbling/expansion etc, as it's generally recommended to have the flask less than half full. Perhaps the authors used a 500 mL flask but wrote 250 mL by mistake.

19) In materials and methods section, section 3.2, the compound numbers in the titles can be made bold.

20) Their should be a short description for the POM, DSC and LOI experiments in the materials and method section.

Author Response

Response to Reviewer 2 Comments

Point 1: Line 21. '...mesogenic with smectic A and C phases' should become 'mesogeneic with either smectic A or C phases'

Response 1: Line 21: compounds with alkoxy substituents are mesogenic with either smectic A or C phases.

Point 2: Line 60. 'polymers liquid crystals' should become 'polymer liquid crystals'

Response 2: Line 60:… phosphazene polymer liquid crystals…

Point 3: Line 97.  'which was then reduced to give' should be changed to 'which was then hydrolysed to give', as the conversion of 4 to 5 is the hydrolysis of an ester to an acid, which are at the same oxidation state. 

Response 3: Line 96-97: HCCP with methyl 4-hydroxybenzoate formed hexasubstituted cyclotriphosphazene benzoate, 4 which was then hydrolyzed to give the subsequent benzoic acid, 5.

Point 4: Section 2.2 and 2.3. Here the authors provide a very detailed and solid rationale for their deduction of the chemical structures from the spectral information. Although the information is correct, I feel that this is perhaps 'over-kill', and that a lot of the information here could be moved to the SI and replaced with a sentence like 'the chemical structures of all compounds were consistent with the spectral data'.

Response 4: The repeated explained data was deleted since the data already available on the compact data in Section 3.3. Some of the data were summarized using the sentence 'the chemical structures of all compounds were consistent with the spectral data'.

Point 5: Line 150. surrounding should be replaced by surroundings.

Response 5: Line 147: … moisture from the sample or surroundings.

Point 6: Line 156. 'Inserted into' should be replaced by 'attached to'

Response 6: Line 149-150: The disappearance band at 3320 cm-1 for O-H stretching indicated that 4-nitrophenol was successfully attached to HCCP.

Point 7: Line 160. 'with a different synthesis route' could be changed to 'by a different synthetic route'

Response 7: Line 152: compound 8j was synthesized by a different synthetic route.

Point 8: Line 168. 'which confirming' should be just 'confirming'

Response 8: Line 158-159: Intermediate 10 was reacted with 3i to form compound 8j with the absorption bands at 3217 and 3451 cm-1, confirming that the coupling reaction was successful.

Point 9: Line 396. 'the SmA phase was disappeared' should become 'the SmA phase disappeared.

Response 377: However, the SmA phase disappeared when intermediates 2a-e

Point 10: Line 427. 'Chlorine is polar substituents possessing...' should become 'Chloride is a polar substituent possessing...'

Response 10: Line 408: Chlorine is a polar substituent possessing strong dipole moments ….

Point 11:  Line 562. 2e 13C NMR data, two carbon environments are missing (20 instead of 22). If this is due to resonance overlap, then should write '2 carbon environments not observed due to coincident resonances', or similar. 

Response 11: Line 568-569: two carbon environments not observed due to coincident resonances (added)

Point 12: Line 576. 3a 13C NMR data, one carbon environment is missing (14 instead of 15). If this is due to resonance overlap, then should write 'one carbon environments is not observed due to coincident resonances', or similar.

Response 12: Line 583-584: one carbon environments is not observed due to coincident resonances (added)

Point 13: Line 584. 3b 13C NMR data, one carbon environment is missing (16 instead of 17). If this is due to resonance overlap, then should write 'one carbon environments not observed due to coincident resonances', or similar.

Response 13: Line 592-593: one carbon environments is not observed due to coincident resonances (added)

Point 14: Line 871. 8e 13C NMR data, one carbon environment too many (27 instead of 26).

Response 14: Line 885-887: The 13C NMR data was corrected with 26 carbon only

Point 15: Line 881. 8g. 13C NMR data, one carbon environment is missing (12 instead of 13. If this is due to resonance overlap, then should write 'one carbon environments not observed due to coincident resonances', or similar.

Response 15: 907-908: one carbon environments not observed due to coincident resonances (added)

Point 16: After 1H NMR data for compounds 3f, 5, 6a, 6b, 3g, 6h, 6i, 8f, 8g should include the sentence 'one exchangeable proton was not observed', or similar.

Response 16: Line 626-627 (3f), Line 639 (3g), Line 709 (6a), Line 720-721 (6b), Line 786-787 (6h), Line 796 (6i), Line 895 (8f), Line 905-906 (8g): one carbon environments not observed due to coincident resonances (added)

Point 17: After the 1H NMR data for compound 6g, should include the sentence 'two exchangeable protons were not observed'.

Response 17: Line 776-777: two exchangeable protons were not observed (added)

Point 18: Line 624. In section 3.2.4, the reaction consists of 200 mL solvent + >35 g reagents in a 250 mL flask. Safety-wise, this seems rather full to allow for bubbling/expansion etc, as it's generally recommended to have the flask less than half full. Perhaps the authors used a 500 mL flask but wrote 250 mL by mistake.

Response 18: Line 631-632: A mixture of 1,4-phenylenediamine (19.44 g, 0.18 mol) and 4-nitrobenzoic acid (16.70 g, 0.1 mol) was mixed in 500 mL round bottom flask containing 200 mL of 3% aqueous NaOH.

Point 19: In materials and methods section, section 3.2, the compound numbers in the titles can be made bold.

Response 19: All the compounds number was bold

Point 20: Their should be a short description for the POM, DSC and LOI experiments in the materials and method section

Response 20: Line 485: The short description for the instruments such as FTIR, NMR, CHN elemental analysis, POM, DSC, and LOI was added in section 3.2

Reviewer 4 Report

This manuscript describes synthesis of new hexasubstituted cyclotriphosphazene derivatives and study of the properties of these compounds. The results are presented correctly (the paper is a logical), compounds are correctly characterized. Particularly noteworthy is the extensive synthetic part. Unfortunately, some of the compounds obtained did not have the intended liquid crystal properties, while others had very high phase transition temperatures. Fire retardant tests are interesting.

The word 'reduction' appears several times in the manuscript, but in most cases it is misused. Lines 97, 102, 162, 165, 168, 214 describes other chemical reactions such as hydrolysis, deprotection of functional groups or even coupling reactions. This needs to be improved.

Author Response

Response to Reviewer 3 Comments

Point 1: The word 'reduction' appears several times in the manuscript, but in most cases it is misused. Lines 97, 102, 162, 165, 168, 214 describes other chemical reactions such as hydrolysis, deprotection of functional groups or even coupling reactions. This needs to be improved.

Response 1:

  1. Line 97: Reaction of hexachlorocyclotriphosphazene, HCCP with methyl 4-hydroxybenzoate formed hexasubstituted cyclotriphosphazene benzoate, 4 which was then hydrolyzed to give the subsequent benzoic acid, 5.
  2. Line 102: Intermediate 9 was produced from the reaction of HCCP with 4-acetamidophenol, and then deprotected to form intermediate 10.
  3. Line 162: The substitution reaction of HCCP with 4-acetamidophenol in a basic solution formed hexasubstituted intermediate, 9, which was deprotected to give intermediate 10 (Line 154)
  4. Line 168: Intermediate 10 was reacted with 3i to form compound 8j with the absorption bands at 3217 and 3451 cm-1, confirming that the coupling reactions was successful. (Line 159)
  5. Line 214: The deprotection of 9 to 10 displayed the appearance …. (Line 197)